# Investigating the shared genetic architecture between multiple sclerosis and inflammatory bowel diseases

Yuanhao Yang [1,2], Hannah Musco[1], Steve Simpson-Yap[3,4], Zhihong Zhu[2,5], Ying Wang[1,2], Xin Lin[3], Jiawei Zhang[6], Bruce Taylor [3,7✉], Jacob Gratten [1,2,7✉] & Yuan Zhou [3,7✉]

An epidemiological association between multiple sclerosis (MS) and inflammatory bowel disease (IBD) is well established, but whether this reflects a shared genetic aetiology, and whether consistent genetic relationships exist between MS and the two predominant IBD subtypes, ulcerative colitis (UC) and Crohn's disease (CD), remains unclear. Here, we use large-scale genome-wide association study summary data to investigate the shared genetic architecture between MS and IBD overall and UC and CD independently. We find a significantly greater genetic correlation between MS and UC than between MS and CD, and identify three SNPs shared between MS and IBD (rs13428812), UC (rs116555563) and CD (rs13428812, rs9977672) in cross-trait meta-analyses. We find suggestive evidence for a causal effect of MS on UC and IBD using Mendelian randomization, but no or weak and inconsistent evidence for a causal effect of IBD or UC on MS. We observe largely consistent patterns of tissue-specific heritability enrichment for MS and IBDs in lung, spleen, whole blood and small intestine, and identify cell-type-specific enrichment for MS and IBDs in CD4+ T cells in lung and CD8+ cytotoxic T cells in lung and spleen. Our study sheds light on the biological basis of comorbidity between MS and IBD.

[1] Mater Research, Translational Research Institute, Brisbane, QLD, Australia. [2] Institute for Molecular Bioscience, The University of Queensland, Brisbane, QLD, Australia. [3] Menzies Institute for Medical Research, University of Tasmania, Hobart, TAS, Australia. [4] Neuroepidemiology Unit, Melbourne School of Population & Global Health, The University of Melbourne, Melbourne, VIC, Australia. [5] National Centre for Register-based Research, Aarhus University, Aarhus, Denmark. [6] Department of General Surgery, the First Affiliated Hospital of Anhui Medical University, Hefei, China. [7] These authors jointly supervised this work: Bruce Taylor, Jacob Gratten, Yuan Zhou. ✉email: bruce.taylor@utas.edu.au; jacob.gratten@mater.uq.edu.au; yuan.zhou@utas.edu.au

Multiple sclerosis (MS) is a complex autoimmune disease of the central nervous system (CNS) manifested by inflammatory demyelination and subsequent neurodegeneration[1]. Inflammatory bowel disease (IBD) is characterised by chronic inflammation of the gastrointestinal (GI) tract, and encompasses both ulcerative colitis (UC; inflammation predominantly in the large intestine and rectum, occasionally in the terminal ileum) and Crohn's disease (CD; inflammation in any part of the GI tract)[2]. Evidence for reciprocal comorbidity of MS and IBD has grown in recent years[3–5]. For example, a large meta-analysis[6] with over one million participants from MS and IBD registries found that MS was associated with a 55% increased risk of IBD, and reciprocally, that IBD patients had a 53% increased risk of MS. No difference in MS prevalence between patients with UC or CD was detected in that study, but others[7,8] have reported greater risk of MS in UC cases, and vice versa, compared to those with CD.

Both MS and IBD are moderately heritable, with estimated liability-scale single nucleotide polymorphism (SNP) heritability of 19%[9] for MS and approximately 25% for IBD (27% for UC and 21% for CD)[10]. Large-scale case-control genome-wide association studies (GWASs) for MS, IBD (case samples including UC and CD), UC and CD have identified hundreds of variants conferring risk for each disease[9,11,12], including some shared risk loci (e.g. IL7R[13,14] and IL2RA[13,15]). These findings suggest that MS may have partially shared genetic risk with UC and CD, but the magnitude of the genetic overlap remains unclear, as does the question of whether any genetic overlap reflects pleiotropy or causality. Interestingly, previous studies[16,17] have reported evidence that MS may share different genetic factors with UC as opposed to CD. For example, MS is genetically more similar to UC than CD in relation to the major histocompatibility complex (MHC) region[18]. Prior studies have also revealed multiple tissues (e.g. lung, spleen, peripheral blood) enriched for SNP heritability of MS, UC and CD (e.g. Finucane et al.[19], IMSGC et al.[9]), although further investigation is needed to determine if this shared enrichment reflects the involvement of the same versus distinct cell types across diseases. Addressing these questions could help to gain a deeper understanding of the biological mechanisms underlying comorbid MS and IBD.

An important issue that clinicians face in neuroimmunology and gastroenterology clinics is how to treat patients with both MS and IBD. For example, it has been reported that the cytokine, interferon-β, used to treat MS can increase the severity of IBD symptoms[20] (although findings are conflicting with some evidence suggesting that interferon treatment may improve symptoms in some IBD patients[21]). Additionally, others have found that TNF-α antagonists that are effective for IBD can worsen the clinical course of MS[22]. These important and not uncommon clinical questions may be helped by an improved understanding of genetic relationships between MS and comorbid IBD that would lead to safer and more effective interventions for both diseases individually and when they occur together.

In this study, we use large-scale GWAS summary data to examine genetic correlations and potential causality between MS and each of IBD, UC and CD. We perform cross-trait GWAS meta-analyses between MS and IBD, UC and CD, and identify genetic risk variants not previously associated with the individual traits. We integrate GWAS summary data with tissue and cell-type-specific gene expression data to determine if SNP heritability for MS and each of IBD, UC and CD are enriched in the same as opposed to distinct tissues and cell types, and we use Summary-data-based Mendelian randomisation (SMR)[23] to identify putative functional genes shared between diseases. A flowchart of our analysis strategy is provided in Fig. S1.

## Results

**Genetic correlations between MS and IBDs.** We first applied stratified linkage disequilibrium (LD) score regression (S-LDSC)[24] with the baseline-LD model[25] to estimate the liability-scale SNP heritability for MS and each of IBD, UC and CD. Consistent with the literature[9,10], the liability-scale SNP heritability (without constrained intercept) was 16% (95% confidence interval [CI] =13–18%) for MS, 19% (95% CI = 16–23%) for IBD, 17% (95% CI = 13–21%) for UC, and 26% (95% CI = 21–31%) for CD. We then used bivariate LDSC to estimate genetic correlations between MS and each of IBD, UC and CD. The genetic correlation (without constrained intercept) between MS and UC ($r_g = 0.33$, $p = 1.66 \times 10^{-13}$) was roughly twice that between MS and CD ($r_g = 0.16$, $p = 2.40 \times 10^{-3}$), which was significant based on Fisher's $Z$-transformation method ($Z$-score=2.39; $p = 0.02$). As expected, the MS-IBD estimate was intermediate between these values ($r_g = 0.28$, $p = 2.01 \times 10^{-10}$), reflecting that the IBD GWAS case sample is comprised of both UC and CD patients (Fig. 1). The intercept of genetic covariance between MS and IBD (or UC or CD) was estimated at around 0.1, indicating mild sample overlap between MS and IBD (or UC or CD). For comparison, the genetic correlation between UC and CD was 0.70 ($p = 2.05 \times 10^{-47}$). These estimates were slightly weaker after constraining the LDSC intercept on the assumption of no population stratification, but nonetheless all remained Bonferroni significant ($p < 1.25 \times 10^{-2}$, see Table S1).

**Local genetic correlations between MS and IBDs.** We used the ρ-HESS (Heritability Estimation from Summary Statistics) method[26] to evaluate local genetic correlations across the genome between MS and each of IBD, UC and CD. In each of the three pairwise comparisons (MS-IBD, MS-UC, MS-CD), there was no evidence for a difference in the average local genetic correlation in regions harbouring MS-specific loci versus IBD-, UC- and CD-specific loci (Fig. 2). Additionally, local genetic correlations in disease-specific loci (e.g. MS-specific and IBD-specific loci for the MS-IBD comparison) were all largely consistent with the genome-wide $r_g$ estimates from bivariate LDSC. These results suggest that MS and IBD (or UC or CD) are likely correlated due to sharing of genetic variation across the entire genome rather than in specific genomic regions. Significant local genetic correlations were identified in the five MHC regions on chromosome 6 for MS-UC and MS-IBD, but not MS-CD, with the caveat that some of the latter estimates may be unreliable due to non-significant local SNP heritability estimates (Table S2; Figs. S2–4).

**Identification of risk SNPs from cross-trait GWAS meta-analysis of MS and IBD.** Based on evidence for significant genetic correlations between MS and each of IBD, UC and CD, we performed cross-trait meta-analyses to identify risk SNPs underlying the joint phenotypes MS-IBD, MS-UC and MS-CD. We used two complementary approaches—MTAG (Multi-Trait Analysis of GWAS)[27] and CPASSOC (Cross Phenotype Association)[28]—and conservatively prioritised only those SNPs surpassing genome-wide significance ($p < 5 \times 10^{-8}$) using both methods. After excluding SNPs that were genome-wide significant in the respective single-trait GWAS (IMSGC GWAS discovery cohort [14,802 cases, 26,703 controls][9]; IBD[12]; UC[12]; CD[12]) or the IMSGC GWAS meta-analysis of MS (discovery + replicate cohorts [47,429 cases, 68,374 controls]; $N = 200$ non-MHC genome-wide significant SNPs)[9] or that were in LD (LD $r^2 \geq 0.05$) with any of these previously reported genome-wide significant SNPs, we identified one SNP (rs13428812) associated with the joint phenotype MS-IBD, which was also significant in the MS-CD cross-trait GWAS (Table S3). A further two SNPs

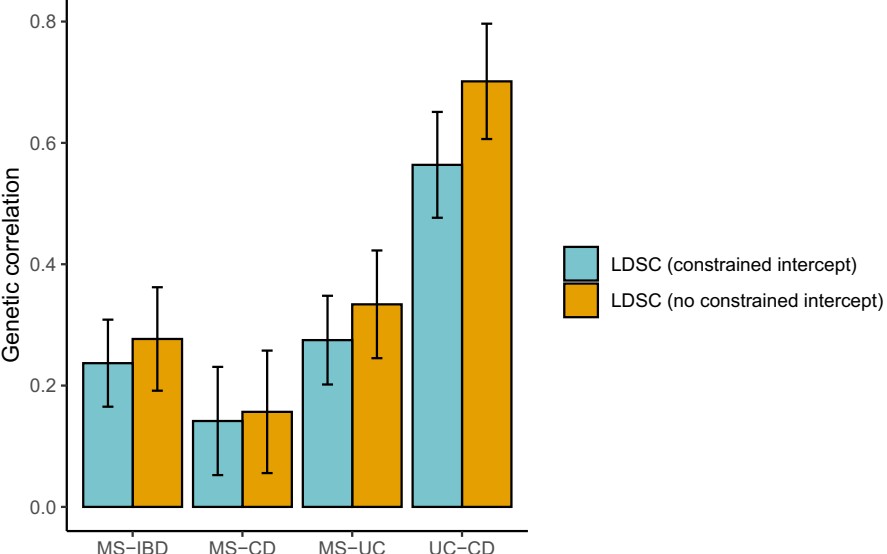

**Fig. 1 Summary of pairwise genetic correlations estimated using linkage disequilibrium score regression (LDSC) with and without constrained intercept.** Bars represent the point estimates of genetic correlation for each disease pair. Error bars represent the 95% confidence intervals (CIs) of the estimated genetic correlations. See Table S1 for complete LDSC results. MS: multiple sclerosis. IBD: inflammatory bowel disease. CD: Crohn's disease. UC: ulcerative colitis. Source data are provided with this paper.

were uniquely associated in the cross-trait GWAS meta-analyses of MS-UC (rs116555563) and MS-CD (rs9977672), respectively. The maxFDR (i.e. the upper bound for the false discovery rate [FDR]) values for MTAG analyses of MS and each of IBD, UC, and CD were roughly $4.55 \times 10^{-7}$. Additionally, MTAG results were highly consistent with those generated by CPASSOC (Figure S5), suggesting any bias due to violation of MTAG assumptions is likely to be negligible.

**Suggestive but inconclusive evidence for causality between MS and UC but not CD.** Next, we used bi-directional Mendelian randomisation (MR) to explore if genetic overlap between MS and each of IBD, UC and CD was consistent with pleiotropy (as we would intuitively expect) or the presence of one or more causal relationships. We applied multiple ($N = 6$) bi-directional MR methods to each pair of phenotypes (MS-IBD, MS-UC, MS-CD), with the rationale that robust relationships would exhibit consistent and statistically significant results across different methods, including CAUSE (Causal Analysis Using Summary Effect estimates)[29], which is the only method capable of distinguishing causality from both correlated and uncorrelated pleiotropy. We found consistent evidence for a causal effect of MS on UC and IBD using five of six MR methods (Bonferroni threshold $p \leq 8.3 \times 10^{-3}$, based on three bi-directional comparisons), but CAUSE could not distinguish a model of causality from correlated pleiotropy for either MS-UC ($p = 0.16$) or MS-IBD ($p = 0.03$; Table S6). In the reverse analyses, there was no or weak and inconsistent evidence for a causal effect of either IBD or UC on MS, and the same was true in bidirectional analyses of MS and CD (Fig. 3, Tables S4, S6). We repeated our analyses with the MHC region excluded, with generally weaker evidence for a causal effect of MS on UC but stronger evidence for a causal effect of MS on CD (Tables S5, S7, Fig. S6).

**Tissue-level SNP heritability enrichment in MS, IBD, UC and CD.** We used S-LDSC[24] to evaluate tissue-level enrichment of SNP heritability for MS, IBD, UC and CD, using Genotype-Tissue Expression (GTEx) data for 37 tissues. We identified Bonferroni- ($p < \sim 3 \times 10^{-4}$) or FDR- ($p < \sim 5 \times 10^{-3}$) significant SNP

heritability enrichment for MS, IBD, UC and CD in each of lung, spleen, whole blood and small intestine–terminal ileum (with the exception of CD), after adjusting for the baseline model and the set of all genes (Fig. 4). Additionally, UC but not MS or CD exhibited Bonferroni-significant enrichment in colon (Figs. S7–9). The magnitude of SNP heritability enrichment in these immune-related tissues ranged from 2.41 to 3.43 and was largely similar in each disease (Supplementary Data 2). The enrichment correlations among MS, UC and CD were relatively high and similar for each trait pair, with estimates ranging from 0.80 to 0.85 (Table S8).

Next, to explore the possibility that sharing of heritability enrichments in immune-related tissues across diseases was due to overlap of highly expressed genes in those tissues, we performed a series of conditional S-LDSC analyses in which—for each focal disease and tissue—we adjusted for the set of genes highly expressed in other Bonferroni- or FDR-significant tissues, in addition to the baseline model and the set of all genes (Supplementary Data 5). In these conditional analyses, the SNP heritability enrichment in all four tissues remained significant for both MS and IBD. Interestingly, the only significant tissue for CD in the conditional analyses was lung, whereas for UC we observed significant enrichment in spleen and small intestine–terminal ileum. These results were generally stronger in more stringent conditional S-LDSC analyses adjusting for all non-focal tissues (Supplementary Data 5, Fig. S39).

**Cell type-level SNP heritability enrichment in MS, IBD, UC and CD.** We extended S-LDSC to investigate cell-type-specific SNP heritability enrichment for MS, IBD, UC and CD in lung, small intestine–terminal ileum, spleen and peripheral blood, using publicly available single-cell RNA sequencing (scRNA-seq) data for a total of 84 cell types in these tissues. We identified FDR-significant ($p < \sim 5 \times 10^{-3}$) enrichment for all four diseases in CD4[+] T cells in lung, and for MS, IBD and CD in CD8[+] cytotoxic T cells in both lung and spleen, and regulatory T cells in lung (Fig. 5, Figs. S16–18, S21–23, S26–28, S31–33, Supplementary Data 4). We also observed significant enrichment for MS and UC in enterocyte progenitors and early enterocyte progenitors in

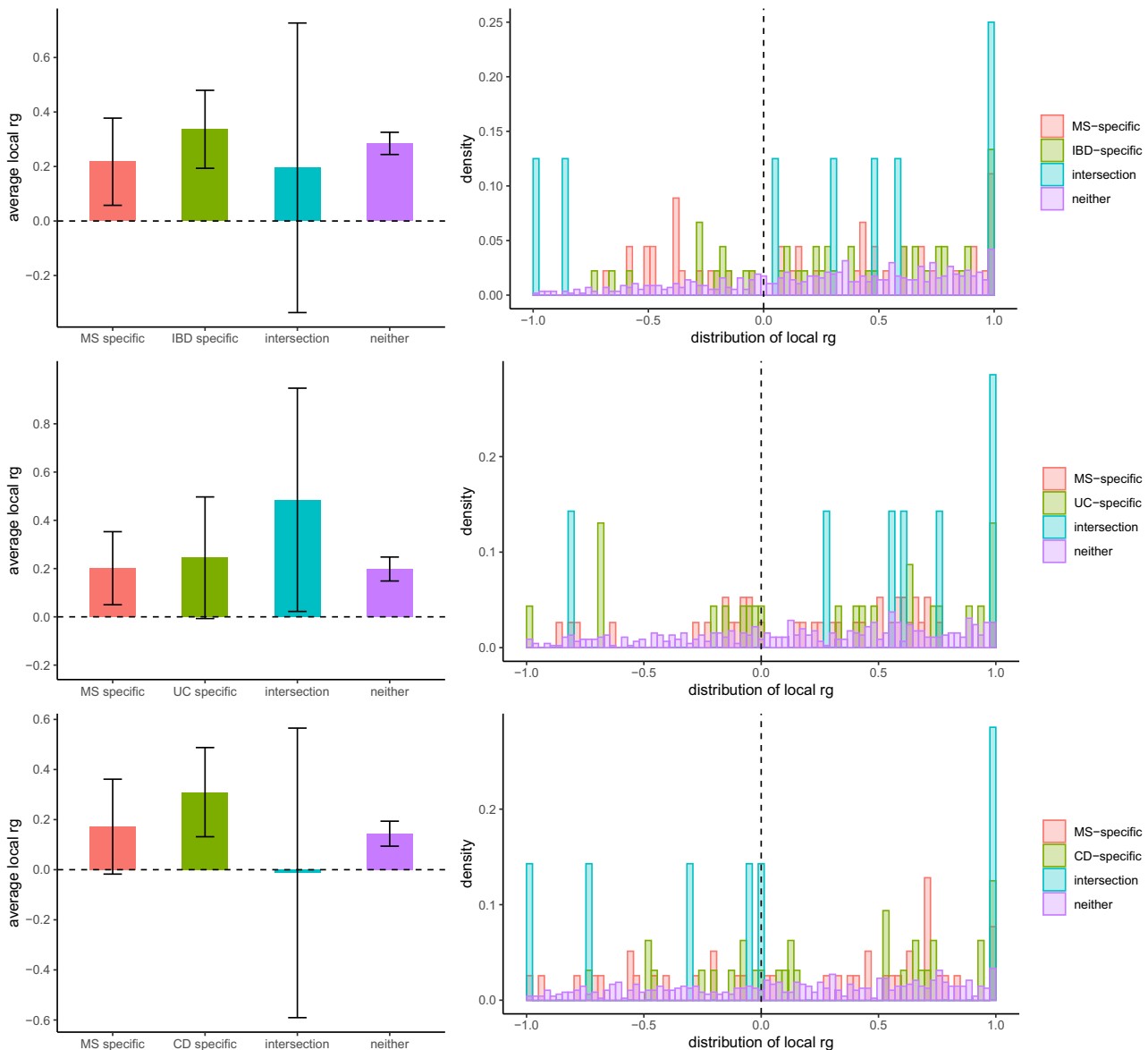

**Fig. 2 Local genetic correlations ($r_g$) between multiple sclerosis (MS) and inflammatory bowel disease (IBD; top), ulcerative colitis (UC; middle) and Crohn's disease (CD; bottom), respectively.** Left: average local $r_g$ estimates for each disease pair in regions harbouring disease-specific risk variants ($p < 5 \times 10^{-8}$), regions harbouring shared risk variants ("intersection") and all other regions ("neither"). Local genetic correlations with estimates less than −1 or greater than 1 were forced to −1 or 1, respectively. Error bars represent the 95% confidence intervals (CIs), calculated using a jack-knife method. Right: density distribution of local $r_g$ estimates for each disease pair in disease-specific regions (red, green), intersection regions (blue) and other (purple) regions. For MS-IBD, 45, 45, 8, and 572 regions were included in the 'MS-specific', 'IBD-specific', 'intersection', and 'neither' categories; for MS-UC, 38, 23, 7, and 456 regions were included in the 'MS-specific', 'UC-specific', 'intersection', and 'neither' categories; for MS-CD, 39, 32, 7, and 480 regions were included in the 'MS-specific', 'CD-specific', 'intersection', and 'neither' categories. Source data are provided with this paper.

the small intestine-terminal ileum, respectively. Cell type-specific SNP heritability enrichment for individual diseases was observed for MS in naïve B cells and dividing T cells in lung, B hypermutation cells and B/T doublets in spleen, and transitional amplifying cells in small intestine. For IBD, we observed enrichment in CD8+ gamma/delta cells in spleen and CD56+ natural killer (NK) cells in peripheral blood. We also observed shared SNP heritability enrichment for IBD, UC and/or CD in a number of dendritic cell types and NK cells in lung. Cell-type-specific heritability enrichments in MS were strongly and significantly correlated with those in IBD, UC and CD in lung ($r > 0.67$, $p < 8.5 \times 10^{-5}$), whereas there was no evidence for a correlation in spleen and small intestine–terminal ileum

(Table S8). Correlations for MS with IBD, UC and CD in peripheral blood mononuclear cells (PBMC) were also high (~0.6), but only marginally significant due to the modest number of cell types in this tissue ($N = 11$). For comparison, the correlation of cell-type-specific heritability enrichments between UC and CD was ~0.75 in lung and spleen, and ~0.55 in PBMC and small intestine–terminal ileum.

Next, we performed conditional S-LDSC analyses to determine in any cell type-specific enrichments survived adjustment for other FDR-significant cell types in the same tissue. In these conditional analyses, CD4+ T cells in lung remained significant for all four diseases. Additionally, CD8+ cytotoxic T cells remained significant in lung and spleen for MS

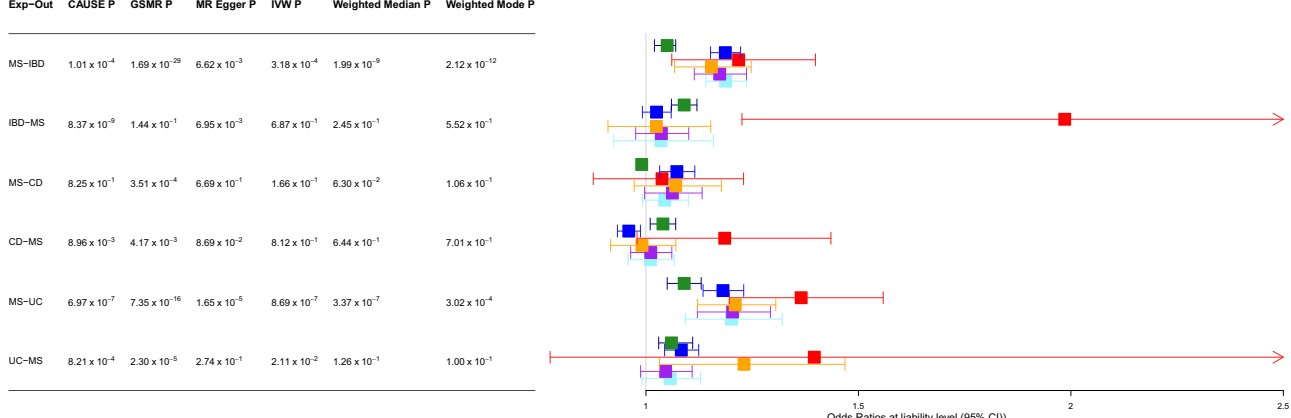

**Fig. 3 Summary of bi-directional Mendelian Randomisation (MR) analyses between multiple sclerosis (MS) and each of inflammatory bowel disease (IBD), ulcerative colitis (UC) and Crohn's disease (CD).** Green: Causal Analysis Using Summary Effect estimates (CAUSE); dark blue: Generalised Summary-data-based Mendelian Randomisation (GSMR); red: MR-Egger; orange: inverse variance weighting (IVW); purple: weighted median; light blue: weighted mode. Error bars represent the 95% confidence intervals (CIs) for the associated MR point estimates. See Tables S4, S6 for complete details of the MR analyses. Source data are provided with this paper.

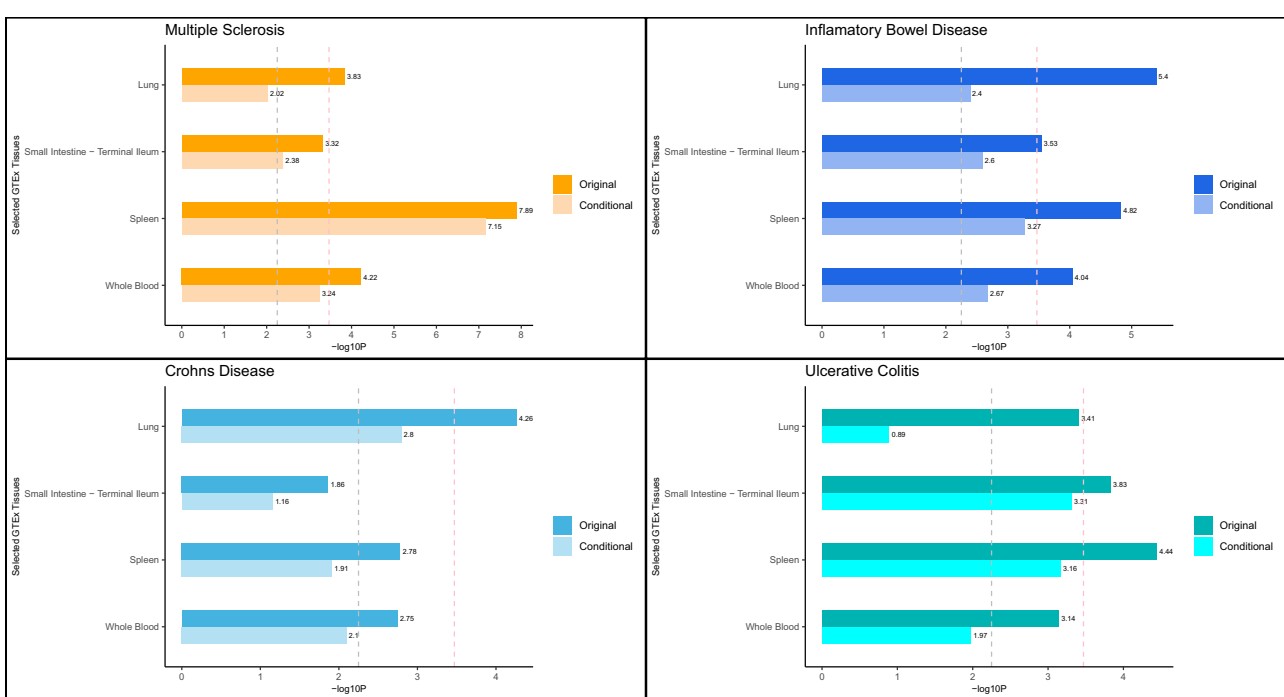

**Fig. 4 Tissue type-specific enrichment of single nucleotide polymorphism (SNP) heritability for multiple sclerosis (MS), inflammatory bowel disease (IBD), ulcerative colitis (UC) and Crohn's disease (CD) in immune tissues estimated using stratified linkage disequilibrium score regression (S-LDSC).** Negative log10 $p$-values of coefficient $Z$-scores for each individual test (two-tailed $Z$-test) are displayed on the $x$ axis. The grey and pink dotted lines represent the FDR threshold <5% and Bonferroni corrected threshold for multiple comparisons, respectively. Original indicates results from S-LDSC analyses adjusted for the baseline model and the set of all genes. Conditional indicates results from conditional S-LDSC analyses that additionally adjusted for the set of genes specifically expressed in the three non-focal tissues (e.g. small intestine–terminal ileum, lung and whole blood in analyses of spleen). Source data are provided with this paper.

and CD, as did enterocyte progenitors and early enterocyte progenitors for MS and UC, respectively (Supplementary Data 6). Another four cell types remained significant (dividing T cells in lung for MS; activated dendritic cells in lung for IBD and UC; CD56$^+$ NK cells in peripheral blood for IBD), but none were shared between MS and IBD, UC or CD. These results were generally stronger in more stringent conditional S-LDSC analyses adjusting for all non-focal cell-types (Fig. S40, Supplementary Data 6).

**Identification of shared functional genes for MS and IBDs.** We used SMR to identify putative functional genes underlying MS and each of IBD, UC and CD, by jointly analysing GWAS summary data for MS, IBD, UC and CD and eQTL summary data from eQTLGen (whole blood)[30] and GTEx (lung, small intestine–terminal ileum, spleen)[31]. We identified 210 non-MHC genome-wide significant associations ($p_{SMR} < 5.32 \times 10^{-7}$), of which 59 (representing 41 unique genes) survived the HEIDI (HEterogeneity In Dependent Instrument)-outlier test

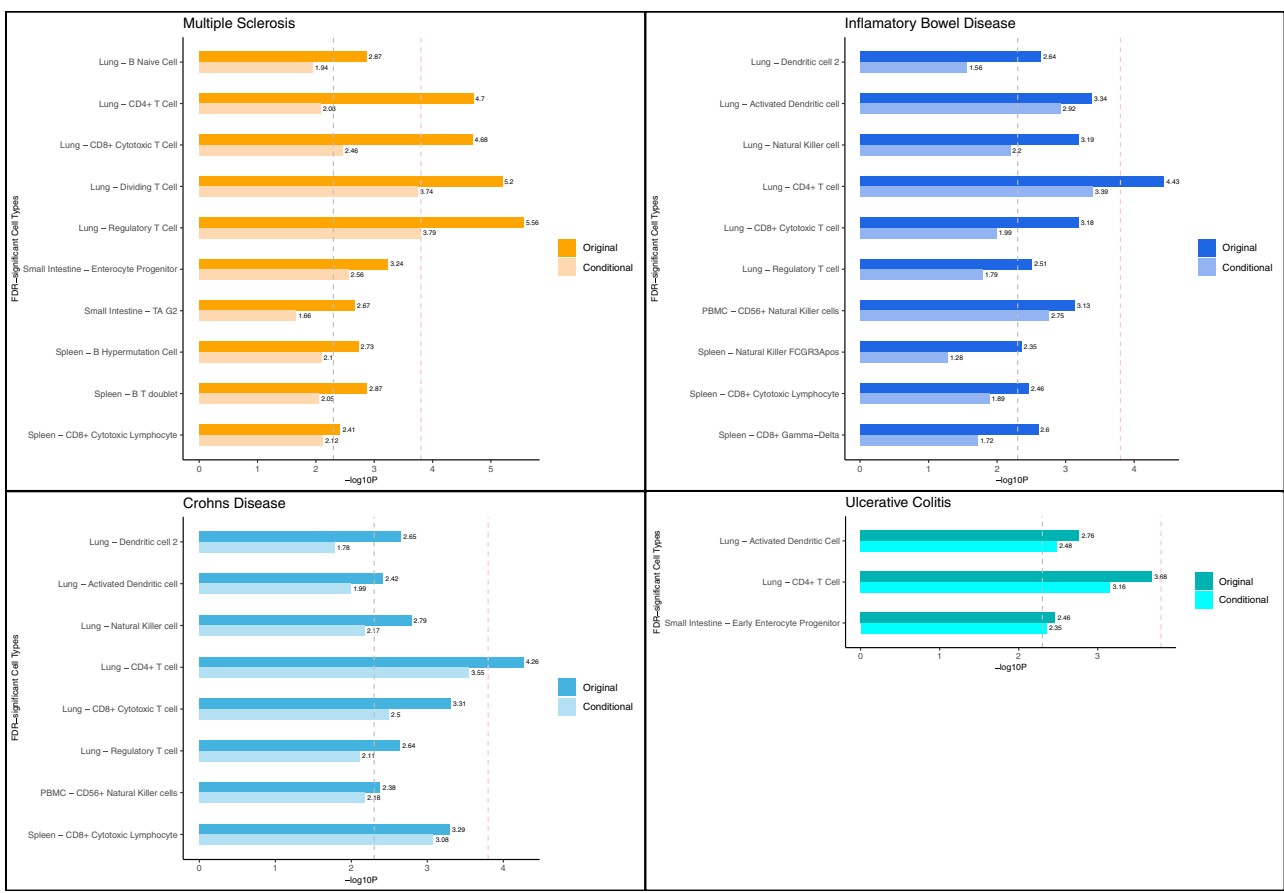

**Fig. 5 Cell-type-specific enrichment of single nucleotide polymorphism (SNP) heritability for multiple sclerosis (MS), inflammatory bowel disease (IBD), ulcerative colitis (UC) and Crohn's disease (CD) in immune tissues estimated using stratified linkage disequilibrium score regression (S-LDSC).** Cell types are included if they showed FDR-significant enrichments in at least one disease. Negative log10 *p*-values of coefficient *Z*-scores for each individual test (two-tailed *Z*-test) are displayed on the *x*-axis. The grey and pink dotted lines represent the FDR threshold <5% and Bonferroni corrected threshold for multiple comparisons, respectively. Original indicates results from S-LDSC analyses adjusted for the baseline model and the set of all genes. Conditional indicates results from conditional S-LDSC adjusted for the baseline model, the set of all genes, the set of genes specifically expressed in the three non-focal tissues (e.g. small intestine–terminal ileum, lung and whole blood in analyses of spleen) and the set of genes highly expressed in other FDR-significant cell types of the same tissue and same disease (e.g. spleen: B/T doublet and spleen: CD8+ cytotoxic lymphocytes in analyses of spleen: B hypermutation cells). Source data are provided with this paper.

(Figs. S41–44, Supplementary Data 11). Among these 41 genes, the only gene shared by MS and one or more of IBD, UC or CD was *GPR25*, which was Bonferroni significant for MS ($p_{SMR} = 1.18 \times 10^{-9}$, $p_{HEIDI} = 0.21$), IBD ($p_{SMR} = 2.91 \times 10^{-10}$, $p_{HEIDI} = 0.12$) and UC ($p_{SMR} = 4.98 \times 10^{-8}$, $p_{HEIDI} = 0.63$), but not CD ($p_{SMR} = 4.82 \times 10^{-5}$, $p_{HEIDI} = 0.63$). The remaining genes were associated with either MS ($N = 25$) or one or more of IBD ($N = 9$), UC ($N = 3$) and CD ($N = 10$), with the majority identified in whole blood ($N = 34$) rather than lung ($N = 9$), spleen ($N = 6$) or small intestine–terminal ileum ($N = 2$). Identified genes included two genes for MS (antisense gene *LL22NC03-86G7.1*, pseudogene *AC100854.1*), one gene for IBD and CD (pseudogene *AL133458.1*), and numerous previously reported genes for MS (e.g. *CD40*[32], *MMEL1*[33]) and UC and/or CD (e.g. *CARD9*[34], *GSDMB*[35], and *ERAP2*[36]). In addition, a total of 253 genes from the MHC region were observed to be associated with MS and IBDs, but only 14 (representing 9 unique genes) passed the HEIDI-outlier test (Supplementary Data 12) and none were shared by MS and IBD, UC or CD. We further identified one previously reported gene (i.e. *DNMT3A*[37,38], Table S9) that were associated with cross-trait meta-analysis phenotypes of MS-IBD or MS-CD and surpassed HEIDI-outlier test.

## Discussion

By leveraging large GWAS datasets and tissue and cell-type-specific expression data, our study provided insights into the shared genetic architecture underlying MS and each of UC and CD.

We identified a significantly stronger genetic correlation between MS and UC than between MS and CD, suggesting that genetic factors make a stronger contribution to comorbidity of MS and UC than MS and CD. Notwithstanding that both genetic and environmental factors may contribute to disease comorbidity, our findings are more consistent with epidemiological reports of stronger comorbidity between MS and UC, as opposed to CD (e.g. Bernstein et al.[7], Gupta et al.[8]), than with studies reporting no detectable difference in prevalence of MS between patients with UC and CD and vice versa (e.g. Kosmidou et al.[6]). However, we note that Kosmidou et al.[6] ($N\sim1$ Million) is a much larger study compared to either Bernstein et al.[7] ($N\sim8000$) or Gupta et al.[8] ($N\sim10,000$).

Analysis of local genetic correlations between MS and each of IBD, UC and CD were largely compatible with the MR analyses (see below), insomuch as there was no evidence for a causal effect of MS on IBD, UC or CD, or vice versa. The observation that

regional $r_g$ estimates were similar to global $r_g$ estimates from LDSC is consistent with the idea that many genetic variants across the genome have pleiotropic effects on these traits. In relation to the MHC region, we observed significant local genetic correlations between MS and UC, but not CD. These results are consistent with prior evidence[18,39,40] for a stronger shared contribution of the MHC to MS and UC, compared to MS and CD, although the picture is complex because some local genetic correlations for MS-UC were positive whereas others were negative. MR analyses excluding the MHC region were largely consistent with this, with generally weaker evidence for causal effects of MS on UC (and IBD). In addition, we cannot effectively distinguish potentially horizontally pleiotropic SNPs from SNPs with causal effects in the MHC region, because of the complex LD structure in this region of the genome.

Cross-trait GWAS meta-analyses identified three SNPs shared between MS and IBD, UC, and CD, which have not been reported previously. Notably, all three SNPs showed the consistent direction of effect between the cross-trait GWAS meta-analyses and the component single trait GWAS's (Table S3). Moreover, the SNP shared by MS-IBD and MS-CD (i.e. rs13428812) also showed the same direction of effect in both cross-trait meta-analyses (Table S3). The results suggest that these SNPs are likely involved in regulating common pathways shared between MS and IBDs.

MR analyses suggested that the genetic correlation between MS and CD is consistent with horizontal pleiotropy, whereas the evidence was inconclusive with respect to the genetic relationship between MS and UC (and IBD). A consistent causal effect of MS on UC (and IBD) was inferred using five of six MR methods with the exception of CAUSE. Since CAUSE is the only MR method capable of distinguishing causality from correlated pleiotropy, we could not further rule out the possibility of horizontal pleiotropy, which is a limitation of our analyses (see below). We note that inference on causality from individual MR methods varied dramatically (e.g. for MS and IBD, GSMR inferred a causal effect of MS on IBD and no effect of IBD on MS, whereas CAUSE inferred a causal effect of IBD on MS), highlighting the importance of considering multiple methods in MR analyses. Larger and more powerful GWAS for MS, and IBDs will be needed to definitively establish (or rule out) the existence of causal relationships between these diseases.

We replicated previous reports of significant SNP heritability enrichment for each of MS[9,19], IBD[19], UC[19], and CD[19] in multiple immune system-related tissues, including lung, spleen and whole blood. Additionally, we identified heritability enrichment for MS, IBD and UC (but not CD) in the small intestine–terminal ileum. In conditional analyses adjusting for genes expressed in non-focal tissues, we found that enrichment in lung was significant in MS, IBD and CD, but not UC, whereas enrichment in spleen and small intestine was significant in MS, IBD and UC, but not CD, suggesting a distinct shared aetiology between MS-UC and MS-CD in different tissues. We attribute the discovery of this tissue-level association to the availability of more powerful GWAS summary statistics. In the case of MS, this is due to a larger sample size (i.e. total sample of 41,505 in this study compared to 17,698 for Finucane et al.[19]), whereas for IBD, UC and CD it is due to more sophisticated statistical methods (i.e. earlier GWAS performed using a meta-analysis of 15 cohorts[11], compared to individual-level bivariate linear mixed-effects model with genetic relatedness matrix as random-effect[12]) yielding more genome-wide significant independent SNPs in comparison to earlier GWAS (i.e. 202, 134 and 165 loci for IBD, UC and CD, compared to 110, 23 and 30, respectively). Alterations in small intestine physiology have been reported to be responsible for triggering both MS and IBDs. For example, pro-inflammatory $T_H17$ (interleukin-17-producing T helper) cells, which are redirected to and regulated by the small intestine[41], have been implicated in the pathogenesis of both MS[42] and IBD[43].

We then extended S-LDSC to the cellular level, identifying a number of additional findings in comparison to Finucane et al.[19] (we also did parallel S-LDSC analyses using the Finucane et al. method, see results in Figs. S14,15 and Supplementary Data 3), who reported seven cell types with significant heritability enrichments in MS, IBD and CD, but not UC, based on analysis of ImmGen data[44] (DC.8-4-11b+.MLN [myeloid cells] in mesenteric lymph nodes, T.4.Pa.BDC [T cells] in pancreas, T.4Mem44h62l.LN [T cells] in subcutaneous lymph nodes) and haematopoietic ATAC-seq data[45] (CD4, CD8, B and NK cells in peripheral blood and bone marrow).

First, we identified SNP heritability enrichments for MS, IBD, UC and CD in CD4+ T cells in lung, all of which remained significant in the conditional S-LDSC analyses. Several CD4+ T cell-related genes have been reported to be involved in the risk of MS, UC and CD[46,47]. For instance, IL23A (Interleukin-23A), which mediates CD4+ T cell function through its receptor IL23R, was reported to be involved in the pathophysiology of MS[48] and IBD[49]. The IFNG gene has also been found to be associated with both MS[50] and IBD[51], through the regulation of Th1 and Th2 cytokines. Both IL23A and IFNG were highly expressed in CD4+ T cells in lung in our analyses.

Second, we found significant SNP heritability enrichments in CD8+ cytotoxic T cells in both lung and spleen as well as regulatory T cells in lung in MS, IBD and CD, but not UC. Interestingly, similar enrichment in these T cells was also observed in UC, but these became non-significant after adjusting for the baseline models, indicating that the enrichment signals from these T cell-specific genes in UC can be explained by pathways associated with the baseline annotations. Notably, the heritability enrichment in CD8+ cytotoxic T cells in lung and spleen for both MS and CD remained significant in the conditional S-LDSC analyses, suggesting a possible role for this cell type in liability to comorbid MS and CD. Several candidate genes involved in the regulation of CD8+ cytotoxic T cells have been implicated in both MS and CD. For example, PTGER4, which encodes the prostaglandin receptor, was found to be involved in susceptibility to both MS[52] and CD[53], possibly through prostaglandin E2 which is relevant to the immune system via regulation of cytokines[54]. Another gene, CXCR6, whose expression is thought to be highly relevant to the immune system via coding of a chemokine receptor protein, has also been reported to be associated with both MS[55] and CD[56]. Both PTGER4 and CXCR6 were highly expressed in CD8+ cytotoxic T cells in our study.

Third, we observed significant heritability enrichment for MS in enterocyte progenitors and for UC in early enterocyte progenitors in the small intestine, in the conditional analyses. These findings are consistent with strong evidence for the involvement of epithelial cells in the small intestine in the pathogenesis of IBD, for example via dysfunction in processing and transmission of antigens to immune cells through the intestinal mucosa[57], and with weaker evidence for their involvement in MS, for instance via intestinal barrier dysfunction[42]. Enrichment in enterocyte progenitors in MS, as opposed to early enterocyte progenitors in UC, may point to related, but not identical pathogenic mechanisms in the small intestine in these diseases, although this may also be a consequence of insufficient study power and/or reliance on small intestine data from mouse, as opposed to human tissues.

Of note, we did not replicate previously reported SNP heritability enrichments in any PBMC cell type for either MS or IBDs. This observation may be explained by differences in the cell type-specific reference data used in our study (i.e. PBMC data)

compared to that in prior papers (e.g. Finucane et al.[19], bone marrow, haematopoiesis ATAC-seq data[58]). We did replicate significant heritability enrichments in several other cell types, including B cells and NK cells, in either MS or IBDs but not both, consistent with distinct pathogenic roles of these cells (compared to CD[+] T cells) in triggering MS and IBDs.

We identified one putatively functional gene *GPR25* using SMR and HEIDI, which was associated with MS, IBD and UC. We also identified several genes for MS (*LL22NC03-86G7.1*, *AC100854.1*) and IBD and CD (lncRNA *AL133458.1*), along with many MS-associated protein-coding genes previously reported by Jacob et al.[59]. *GPR25* is a G protein-coupled receptor that is highly expressed in T cells and NK cells and has been shown to be involved in the risk of MS and IBD[60]. In contrast, little is known about the function of lncRNAs *AL133458.1*, or of antisense gene *LL22NC03-86G7.1* or pseudogene *AC100854.1*, but pseudogenes and non-coding RNAs have previously been hypothesised to make crucial contributions to comorbid MS[61,62] and IBD[63,64].

Notably, our findings suggest that the MHC region likely makes only a modest contribution to the shared genetic architecture underlying MS and IBDs. Evidence supporting this conclusion includes (1) significant local genetic correlations in some LD-independent MHC regions for MS-IBD and MS-UC; (2) modest increases in MR effect size with the inclusion of genes in the MHC region for MS-IBD and MS-UC (Tables S4–7); (3) modest increases in heritability enrichment for MS and each of IBD, UC, and CD, with the inclusion of genes in the MHC region (see details in the Supplementary Note, Supplementary Data 7–10 and Figs. S10–13, S19–20, S24–25, S29–30, S34–38). Nevertheless, quantifying the genetic contribution of the MHC region in disease is challenging and further studies are warranted.

The clinical implications of our findings are multifaceted. Firstly, they help to explain the differential effects of immunomodulating therapies between MS and IBD. Previously researchers have found that type I interferons (including interferon-β, a pivotal MS therapy until recently) have tissue-specific and gene-specific effects that are dependent on a balance of different cytokines[65]. Although the immunomodulatory mechanism for interferon-β therapy in MS is not fully understood, its key role is in presumably inhibiting transmigration of autoreactive T cells into the CNS[66]. Given its tissue-specific effects, interferon-β treatment may unexpectedly lead to immunopathology in other tissues by influencing dendritic cell activation and maturation, and NK cell activity[67]; both of these cell types showed significant heritability enrichments for IBD in our study, potentially explaining the deleterious effect of these medications in some cases of IBD. Similarly, anti-TNF therapy for IBD can be associated with the development of MS-like lesions in the CNS and/or worsening of extant MS, although the mechanism is unclear. Anti-TNF therapy does reduce CD8[+] T cell proliferation, which can result in viral infection or reactivation[68]. In our study, CD8[+] cytotoxic T cells in lung showed significant heritability enrichment for MS while it is well known that elevated EBNA-1 IgG level (due to Epstein-Bar virus infection or reactivation) is an important risk factor for MS development and progression[69]. Therefore, fine tuning current treatments or developing new therapies that are capable of delivering targeted effects in diseases-specific tissues and/or cell types may be a solution to tackle the problem of concomitant worsening of comorbid MS or IBD with some therapies. What is more, our findings can be used as a foundation for future studies to map causal tissues, cells, and genes to gain insights into the pathogenesis of these diseases, which ultimately could lead to novel drug treatments that may improve clinical outcomes.

Our study had several limitations. First, our cross-trait GWAS meta-analysis results may be biased due to violations of the MTAG assumptions of equal SNP heritability for each trait and perfect genetic covariance between traits. However, such impacts were likely negligible since we performed CPASSOC as a sensitivity analysis and observed highly consistent results. Second, with reference to our MR analyses, we note that CAUSE is the only MR method (of the six implemented in our study) capable of distinguishing causality from correlated pleiotropy (i.e. a shared heritable factor affecting both exposure and outcome), and thus the latter may explain the inconsistent inference between CAUSE and the other MR methods in relation to a causal effect of MS on both UC and IBD. Evidence suggests that correlated pleiotropy is common and may frequently contribute to false-positive inference in MR[29]. Further investigations in larger, more powerful datasets will be needed to determine if MS is causal for IBDs. Third, we evaluated tissue and cell type-specific heritability enrichments on the basis of the top 10% of most specific genes, which may neglect influences from other genes with less specific effects. Fourth, we only selected nearby SNPs of the top genes and excluded the SNPs in the MHC region for LD score regression, which may result in underestimation of genetic correlations between MS and IBDs as well as heritability enrichments per tissue and cell for MS and IBDs.

In summary, our study revealed significantly stronger genetic overlap between MS and UC compared to MS and CD, but evidence on whether this represents a causal effect in relation to MS and UC was inconclusive. We identified three SNPs shared between MS and IBD (or UC or CD), none of which was genome-wide significant in the single-trait GWAS, and one candidate gene (*GPR25*) significantly implicated in susceptibility to cross-trait MS and IBD (or UC). We revealed evidence for shared SNP heritability enrichment for MS and UC (or IBD) in small intestine–terminal ileum, for MS and CD in CD8[+] cytotoxic T cells in lung and/or spleen and for all four diseases in CD4[+] T cell in lung, all of which remained significant after conditioning on other tissues and cell types. These findings progress our understanding of shared genetic mechanisms underlying both MS and IBDs, and potentially provide points of intervention that may allow the development of new therapies for these common immune disorders.

## Methods

### Study samples

*GWAS dataset for MS*. GWAS summary results for MS were obtained from the International MS Genetics Consortium (IMSGC) meta-analysis of 15 datasets comprising 14,802 MS cases and 26,703 controls of European ancestry[9]. Each dataset was imputed using the 1000 Genomes European panel. SNPs with minor allele frequency (MAF) > 1% were utilised for meta-analysis using a fixed-effects model. As the MAF information was not available in the MS GWAS meta results, we annotated the MAF information based on the European population from the 1000 Genomes panel. Ambiguous SNPs (AT, TA, CG and GC) were excluded and a total of ~6.8 million SNPs were retained for analysis.

*GWAS datasets for IBD, UC and CD*. We obtained publicly available GWAS summary data for UC, CD and IBD, the latter case sample comprising those in both the UC and CD GWAS[12]. We note that UC and CD were the primary focus of our analyses, but we also included IBD, so as to compare the results of our genetic analyses to the epidemiological literature for overlap between MS and IBD, and because the GWAS for IBD has greater power than UC and CD alone. A total of 34,652 participants of European ancestry (12,882 cases and 21,770 controls) were included in the IBD GWAS, from which 27,432 Europeans (6968 cases and 20,464 controls) and 20,883 Europeans (5956 cases and 14,927 controls) were included in the UC and CD GWAS, respectively. Nearly 12 million SNPs (~9.5 million with MAF > 1%) were included in all three GWAS summary statistics, imputed using the 1000 Genomes Europeans as the reference. Genome-wide association analyses for each disease were conducted using PLINK[70], adjusted by principal components. More details about the cohorts and quality control (QC) process are explained in Jostins et al.[11] and Liu et al.[12].

*GTEx data*. GTEx is a public data resource of gene expression in 53 non-diseased human primary tissues[31], including 50 solid tissues (e.g. liver, stomach), including some organs (e.g. brain) with multiple subregions, whole blood and two cell line 'tissues' (e.g. Epstein–Barr virus–transformed lymphocytes). We used normalised

(transcripts per million) GTEx V7 data[71] to assess tissue-specific gene expression. After excluding low-quality individuals ($N = 2$, defined as <100 genes with >1 read per million) and genes ($N = 736$, defined as <4 individuals with >1 read per million), we retained data on 53 tissues from a total of 751 individuals, with an average of 220 samples per tissue type. In addition, we also downloaded the GTEx V7 expression quantitative trait locus (eQTL) summary data for the downstream analysis.

*scRNA-seq data.* On the basis of evidence for tissue-level SNP heritability enrichment in the GTEx analyses, we obtained scRNA-seq unique molecular identifier (UMI) count matrices from healthy human lung ($N = 57{,}020$ cells)[72], spleen ($N = 94{,}257$ cells)[72] and peripheral blood ($N = 68{,}579$ cells)[73], and mouse small intestine[74] ($N = 7216$ cells). For the latter, we filtered genes with mismatched gene symbols between mouse and human. In our analyses, we used the normalised and quality controlled scRNA-seq data and cell clustering results reported in the primary articles[72–74] and no further QC was conducted. A total of 84 cell types across four tissues were utilised in our study (see Supplementary Data 1), with an average of 2703 cells per cell type.

## Statistical analyses

*LDSC.* We used S-LDSC[24] (Python 2.7) with the baseline-LD model[25] to estimate single trait SNP heritabilities ($h^2_{SNP}$, i.e. the proportion of the phenotypic variance in a trait can be explained by common genetic variants tagged on SNP arrays) for MS, IBD, UC and CD. Baseline-LD model[25] is an extension of S-LDSC[24] that partitions the SNP heritability on the basis of continuous, as opposed to binary annotation sets[25]. We used this approach to estimate $h^2_{SNP}$ rather than univariate LDSC, as the latter may underestimate $h^2_{SNP}$ due to the action of negative selection, which results in SNPs with low levels of LD having higher per-SNP heritability. We reformatted all GWAS summary statistics to the pre-computed LD scores of the 1000 Genomes European reference. SNPs were excluded if they did not intersect with the reference panel, or if they were located in the MHC region (chromosome 6: 28,477,797-33,448,354), had a MAF < 1% or INFO score <0.3. SNP heritability estimates were converted to the liability-scale based on the observed sample prevalence and population prevalence, assuming the latter were 0.3%, 0.54%, 0.29%, and 0.25%[75,76] for MS, IBD, UC and CD, respectively.

We used bivariate LDSC[77] to estimate genetic correlations ($r_g$, i.e. the proportion of genetic variance shared by two traits divided by the square root of the product of their SNP heritability estimates) between MS and each of IBD, UC and CD, as well as between UC and CD. We conducted bivariate LDSC without constraining the intercept and $r_g$ estimates were considered Bonferroni significant if the $p$-value was $<1.25 \times 10^{-2}$ (i.e. $p < \frac{0.05}{4}$). To compare the significance of the difference between two genetic correlations, we calculated Fisher's transformed $Z$-score from $r_g$ using the formula $Z\text{-score}^{78} = \frac{Z_{r_{g1}} - Z_{r_{g2}}}{s}$, where $r_{g1}$ and $r_{g2}$ represent the two genetic correlations, Fisher's transformed $Z_{r_g} = \frac{1}{2} \ln\left(\frac{1+r_g}{1-r_g}\right)$ per correlation, $s = \sqrt{\frac{1}{n_{r_{g1}}-3} + \frac{1}{n_{r_{g2}}-3}}$ of which effect sample size per correlation is estimated by $n_{r_g} = \frac{1-r_g}{s.e._{r_g}^2} + 2$. We then calculated the two-tailed $p$-value from the $Z$-score of a standard normal distribution.

As a sensitivity analysis, we also performed LDSC with the single-trait heritability intercept constrained. Compared to LDSC with unconstrained intercept, constrained intercept LDSC is an approach designed to decrease the standard error of estimates under the assumption of no population stratification, thereby indirectly evaluating the influence of GWAS statistic inflation. However, constrained intercept LDSC may also provide a biased or misleading estimate of heritability and genetic correlation if the intercept is constrained incorrectly.

*Estimation of local genetic correlations using ρ-HESS.* To investigate whether MS shared higher genetic overlap with UC in the local independent genomic region than CD, we applied ρ-HESS[26] (Python 2.7) to evaluate the local genetic correlations (i.e. genetic correlation between traits due to their shared genetic variance at a defined genomic region) between MS and each of IBD, UC, and CD. A total of 1699 default regions that were approximately LD independent with average size of nearly 1.5Mb[79] were checked by ρ-HESS, including five regions in the MHC (i.e. chromosome 6: 28,017,819–28,917,608, 28,917,608–29,737,971, 30,798,168–31,571,218, 31,571,218–32,682,664, and 32,682,664–33,236,497). We performed ρ-HESS to estimate the local SNP heritability per trait and genetic covariance between traits based on the 1000 Genomes Europeans reference of hg19 genome build. Local genetic correlation estimates were then calculated from the local single-trait SNP heritability and local cross-trait genetic covariance estimates.

*Multi-trait analysis of GWAS.* To identify risk SNPs associated with joint phenotypes comprising MS and each of IBD, UC, and CD, we implemented cross-trait meta-analysis of GWAS summary statistics using MTAG[27] (Python 2.7). We used MTAG, rather than standard inverse-variance weighted meta-analyses with trait-specific effect sizes, because this approach can accommodate potential sample overlap between GWAS. We implemented MTAG options that assume equal SNP heritability for each trait and perfect genetic covariance between traits. The upper bound for the false discovery rate ('maxFDR') was calculated to examine the

assumptions on the equal variance–covariance of shared SNP effect sizes underlying the traits. To investigate if violations of the assumptions of equal SNP heritability for each trait and perfect genetic covariance between traits biased our MTAG results, we performed CPASSOC[28] for MS-IBD, MS-UC, and MS-CD as a sensitivity analysis. CPASSOC assumes the presence of heterogeneous effects across traits and estimates the cross-trait statistic $S_{Het}$ and $p$-value through a sample size-weighted meta-analysis of GWAS summary data. We prioritised independent SNPs that were genome-wide significant in the cross-trait meta-analyses (e.g. MS-IBD) using both MTAG and CPASSOC, but not identified in the original single-trait GWAS (e.g. MS or IBD). These independent genome-wide significant SNPs were identified by LD clumping ($r^2 < 0.05$ within 1,000-kb windows) using PLINK v1.9[70], based on the UK Biobank European reference combined imputed by Haplotype Reference Consortium (HRC). We defined cross-trait SNPs of particular interest if they were independent (i.e. LD $r^2 < 0.05$ within 1,000-kb windows using the same reference) from genome-wide significant SNPs in the respective single-trait GWAS (IMSGC GWAS discovery cohort [14,802 cases, 26,703 controls][9]; IBD[12]; UC[12]; CD[12]) or the IMSGC GWAS meta-analysis of MS (discovery + replicate cohorts [47,429 cases, 68,374 controls]; $N = 200$ non-MHC genome-wide significant SNPs; for which we did not have access to the full summary statistics)[9] or that were in LD (LD $r^2 \geq 0.05$) with any of these previously reported genome-wide significant SNPs.

*MR analyses.* We used six MR methods to investigate putative causal relationships between MS and each of IBD, UC and CD: Generalised Summary-data-based Mendelian Randomisation (GSMR)[80], MR-Egger[81], inverse variance weighting (IVW)[82], weighted median[83], weighted mode[84] and CAUSE[29]. We utilised multiple MR methods with different assumptions on the extent and nature of horizontal pleiotropy, which refers to variants with effects on both outcome and exposure through a pathway other than a causal effect. Horizontal pleiotropy can be correlated, if variants affecting both the outcome and exposure do so via a shared heritable factor, or uncorrelated, if variants affect outcome and exposure traits via separate mechanisms. We considered relationships with consistent evidence for causality using all MR methods to be more reliable and noteworthy.

We used the R packages *GSMR*[80] and *TwoSampleMR*[85] to implement five MR methods (GSMR, IVW, MR-Egger, weighted median and weighted mode) with different assumptions about horizontal pleiotropy. Briefly, GSMR assumes no correlated pleiotropy but implements the HEIDI-outlier approach to identify and remove SNPs with evidence for significant uncorrelated pleiotropy. IVW assumes that if uncorrelated pleiotropy is present it has mean zero, so only adding noise to the regression of meta-analysed SNP effects with multiplicative random effects[82]. MR-Egger further allows for the presence of directional (i.e. non-zero mean) uncorrelated pleiotropy and adds an intercept to the IVW regression to exclude confounding from such pleiotropy[81]. Two-sample MR methods capable of accounting for some correlated pleiotropy include the weighted median and the weighted mode. The weighted median measures the weighted median rather than weighted mean of the SNP ratio, which has the ability to identify true causality if ≤50% of the weights are from invalid SNPs[83]. The weighted mode classifies the SNPs into groups according to their estimated causal effects, and assesses evidence for causality using only the largest set of SNPs, which essentially relaxes the assumptions of MR and has the ability to identify the true effect even if a majority of instruments are invalid SNPs[84]. For these five MR methods, independent SNPs (LD clumping $r^2 < 0.05$ within 1000-kb windows using PLINK v1.9[70], according to the UK Biobank European reference combined imputed by HRC and UK10K) with evidence for genome-wide association ($p \leq 5 \times 10^{-8}$) with the 'exposure' trait were used as instrumental variables, and merged with the SNPs from the 'outcome' trait.

We also used a recently published Bayesian-based MR method called CAUSE that accounts for both correlated and uncorrelated pleiotropy[29]. Compared to the other two-sample MR methods, CAUSE further corrects correlated pleiotropy by evaluating the joint distribution of effect sizes from instrumental SNPs, assuming that the 'true' causal effect can influence all instrumental SNPs while correlated pleiotropy only influences a subset of instrumental SNPs. CAUSE improves the power of MR analysis by including a larger number of LD-pruned SNPs (LD $r^2 < 0.10$) with an arbitrary $p \leq 1 \times 10^{-3}$, and provides a model comparison approach to distinguish causality from horizontal pleiotropy.

We implemented bi-directional MR analyses using all six methods to investigate the putative causal effect of MS on each of IBD, UC and CD, and vice versa. Due to the complicated LD patterns in the MHC region, here we performed MR analyses with and without SNPs located within the MHC region, to further investigate the effects of MHC region SNPs on putative causal associations between MS and each of IBD, UC and CD. We applied a stricter LD threshold ($r^2 < 0.001$) when pruning SNPs in the MHC region.

We declared inferred causal relationships to be significant if they showed Bonferroni-corrected $p < 8.33 \times 10^{-3}$ ($=\frac{0.05}{6}$) using all MR methods. For all MR methods, we converted our estimated MR effect size from logit-scale to liability-scale using the formula described by Byrne et al.[86] (i.e. $\text{beta}_{xy[\text{liability}]} = \frac{z_{K_x} K_y (1-K_y)}{z_{K_y} K_x (1-K_x)} \text{beta}_{xy[\text{logit}]}$, where $K_x$ and $K_y$ are the population prevalence of exposure and outcome trait, respectively; and $z_{K_x}$ and $z_{K_y}$ are the height of the Gaussian distribution at the population prevalence threshold for exposure and outcome trait, respectively), assuming the population prevalence for MS, IBD, UC and CD were 0.3%, 0.54%, 0.29%, and

0.25%[75,76], respectively. We then transformed the liability-scale effect size to an odds ratio.

*Tissue and cell-type specific enrichment of SNP heritability. Selection of tissue type- and cell type-specific expressed genes*: We selected genes that were specifically expressed in each tissue and cell type using the method described by Bryois et al.[87]. For GTEx, we followed Bryois et al. in excluding testis and tissues that were non-natural or collected in <100 donors. We then calculated the average gene expression for tissues in the same organ (e.g. colon-sigmoid and colon transverse), with the exception of brain tissues. These filtering criteria reduced the total number of analysed GTEx tissues from 53 to 37. Subsequently, for each tissue and cell type, we excluded non-protein coding genes, genes with duplicated names, genes located in the MHC region, and genes not expressed in any tissue or cell type. We then scaled gene expression to a total of 1 million UMIs per tissue or cell type, and calculated, for each gene, the proportion (ranging from 0 to 1) of total expression across all tissue/cell types that were specific to each tissue/cell type. The top 10% of most specific genes for each tissue and cell type were then selected for downstream analyses.

*Stratified LD score regression:* We first used S-LDSC[24] to investigate whether SNP heritability for MS, IBD, UC and CD was enriched in specific tissues. We then applied S-LDSC to scRNA-seq data to evaluate whether specific cell types in those tissues showed significant heritability enrichment. For each of 37 GTEx tissues and 84 cell types from healthy human lung ($N = 28$), spleen ($N = 30$) and peripheral blood ($N = 11$), and mouse small intestine ($N = 15$; we used mouse small intestine data as a 'proxy' because no large human small intestine data was publicly available), we defined a focal functional category by selecting SNPs located within 100Kb (hg19) of the set of 10% most specific genes and added this to the default baseline model (comprising 52 genomic annotations) and the set of all genes. We evaluated the significance of each SNP heritability enrichment estimate using the *p*-value of the regression coefficient *Z*-score, after adjusting for the baseline model and the set of all genes. Enrichment correlations among MS, UC and CD were calculated by correlating the regression coefficients for GTEx tissues and cell types (by tissues) independently. We adjusted for multiple testing by calculating the Benjamini-Hochberg FDR, accounting for tissues and cell types separately across the diseases.

Additionally, we performed a series of conditional S-LDSC analyses to account for the possibility that gene sets overlap between related tissues and cell types, which may limit our interpretation of estimated tissue- and cell type-specific SNP heritability enrichments. In the case of tissue-level analyses, we conditioned on the baseline model, the set of all genes and the set of genes highly expressed in other FDR-significant tissues. For example, in conditional S-LDSC analyses for lung, we adjusted for the baseline model, the set of all genes and the set of genes highly expressed in small intestine–terminal ileum, spleen and whole blood. In the case of cell type-specific conditional S-LDSC analyses, we additionally conditioned on the set of genes highly expressed in the other FDR-significant cell types of the same tissue, for the focal disease. For example, in the case of B hypermutation cells (spleen) in MS, conditional S-LDSC adjusted for the baseline model, the set of all genes, the set of genes highly expressed in small intestine–terminal ileum, lung and whole blood, and the set of genes highly expressed in B/T doublet cells (spleen) and CD8+ cytotoxic lymphocytes (spleen). For all tissues and cell types, we also conducted gene-set enrichment analysis using MAGMA (Multi-marker Analysis of GenoMic Annotation; see details in the Supplementary Note)[88] with and without genes in the MHC region, as a sensitivity analysis for S-LDSC.

*Summary-data-based Mendelian randomisation.* We used SMR to identify putative functional genes underlying statistical associations for MS, IBD, UC and CD, as well as additional loci identified in cross-trait meta-analyses of MS-IBD, MS-UC and MS-CD, motivated by the question of whether common risk genes underlie MS and IBDs. SMR[23] performs a Mendelian randomisation-equivalent analysis that uses summary statistics from GWAS and eQTL studies to test for an association between gene expression (i.e. exposure) and a target phenotype (i.e. outcome), using genome-wide significant SNPs as instrumental variables. A significant SMR association could be explained by a causal effect (i.e. the causal variant influences disease risk via changes in gene expression), pleiotropy (i.e. the causal variant has pleiotropic effects on gene expression and disease risk) or linkage (i.e. different causal variants exist for gene expression and disease). SMR implements the HEIDI-outlier test to distinguish causality or pleiotropy from linkage, but there is currently no way to distinguish causality from pleiotropy.

We implemented SMR using *cis*-eQTL summary data for whole blood from eQTLGen, a meta-analysis of 14,115 samples[30], and from GTEx V7[31] for other significant tissues identified by S-LDSC. We utilised UK Biobank European reference combined imputed by HRC and UK10K to evaluate LD, and only focused on expression probes with eQTL $p \le 5 \times 10^{-8}$. For MTAG or CPASSOC-based cross-trait phenotypes (e.g. MS-IBD), SMR analyses were restricted to genetic variants of particular interest that defined above. Despite the complicated LD structure in the MHC region, probes located in the region were included in sensitivity analyses due to the importance of the MHC region in susceptibility to MS and IBDs, including for the purpose of comparing SMR associations for MS and IBDs in the MHC region with those in the remainder of the genome. SMR associations due to causality or pleiotropy were declared significant if they surpassed Bonferroni-correction for the total number of eQTLs analysed ($N = 94,624$, $p < 5.28 \times 10^{-7}$) and also passed the HEIDI-outlier test ($p > 0.05$, minimum >10 SNPs).

**Reporting summary**. Further information on research design is available in the Nature Research Reporting Summary linked to this article.

## Data availability

GWAS summary statistics for MS are available by application from https://imsgc.net/?page_id=31. GWAS summary statistics for IBD, UC, and CD are publicly available from https://www.ebi.ac.uk/gwas/publications/26192919. GTEx expression summary data are available from https://gtexportal.org/home/datasets. Summary-level scRNA-seq data are available from https://singlecell.broadinstitute.org/single_cell/study/SCP44/small-intestinal-epithelium for mouse small intestine, https://www.tissuestabilitycellatlas.org/ for human lung and spleen, and https://support.10xgenomics.com/single-cell-gene-expression/datasets for human PBMC. The eQTL summary data for eQTLGen and GTEx are available from https://www.eqtlgen.org/cis-eqtls.html and https://cnsgenomics.com/software/smr/#DataResource. Source data for generating Figs. 1–5 are provided as a Source Data file with this paper. Data for generating supplementary Figures are deposited to Zenodo: https://doi.org/10.5281/zenodo.5289334 Source data are provided with this paper.

## Code availability

Custom R scripts for S-LDSC, ρ-HESS analysis and generating the main figures can be found at https://doi.org/10.5281/zenodo.5289334. Other methods used are as follows: LDSC: https://github.com/bulik/ldsc; PLINK: https://www.cog-genomics.org/plink/1.9; MTAG: https://github.com/JonJala/mtag; CPASSOC: http://hal.case.edu/~xzz10/zhu-web/; CAUSE: https://jean997.github.io/cause/index.html; GSMR: http://cnsgenomics.com/software/gsmr/; TwoSampleMR:https://mrcieu.github.io/TwoSampleMR/;MAGMA: https://ctg.cncr.nl/software/magma; SMR: https://cnsgenomics.com/software/smr/#Overview.

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

## Acknowledgements

The authors acknowledge funding support from the Australian National Health and Medical Research Council (JG: GNT1103418, GNT1127440; YZ: GNT1173155) and the Mater Foundation (J.G., Y.Y., H.M., Y.W.). We also thank IMSGC for providing access to their GWAS summary data. This research was enabled using the UK Biobank Resource under application 12505.

## Author contributions

Y.Y., J.G. and Y.Z. designed the study and wrote the manuscript. Y.Y. performed the primary analyses, with assistance from B.T. and Y.Z. (data preparation), H.M. (S-LDSC), Z.Z. (MR analyses), and Y.W. (MTAG). J.G., B.T. and Y.Z. supervised the study. Y.Y., H.M., S.S.Y., Z.Z., Y.W., X.L., J.Z., B.T., J.G. and Y.Z. contributed to the interpretations of the findings and the critical revision of the manuscript.

## Competing interests

The authors declare no competing interests.
