## [Peer Review File · Nature Communications]

REVIEWER COMMENTS

Reviewer #2 (Remarks to the Author):

In this paper, Yang et al. investigate the shared genetic architecture between multiple sclerosis (MS), and inflammatory bowel diseases (IBD), ulcerative colitis (UC), and Chron's disease (CD). They use state of the arts methods such as LD score regression, HESS, MTAG, numerous mendelian randomization methods, and SMR.

I have two general comments:

- 1) While the paper is really easy to read for statistical geneticists, I believe that this paper would be harder to read for geneticists or physicians interested in the shared architecture of MS, IBD, UC and CD.
- 2) It is hard to understand what are the main conclusions. The abstract states that genetic correlation between MS and UC is twice the one between MS and CD, but this difference does not look significant (Figure 1), while mendelian randomization are pretty neat. Another example is that S-LDSC highlight many shared tissue and cell-type enrichments, but it is not clear if it is due to shared architecture, or high correlation between the corresponding annotations; similarly, only one gene is found shared using SMR.

Here are several presentation and methodological suggestions that could correct these two points:

- (minor) p4 -l14: Numerous papers shown that LDSC under-estimates the h^2 of a trait (Speed et al. 2017 Nat Genet, Gazal et al. 2019 Nat Genet, Hou et al. 2019 Nat Genet), but this can be corrected using S-LDSC with the baseline-LD model (Gazal et al. 2019 Nat Genet, Speed et al. 2020 Nat Genet).
- p4-l19: what is the statistical significance between the genetic correlation between MS and UC and the one between MS and CD? Figure 1 suggests that it is not significant.
- p4-l18: not clear to the reader what is the difference between LDSC with intercept and without intercept.
- p4-l29: not clear to the reader what is the difference between genetic correlation and local genetic correlation.
- p5-l17: not sure to get the title of this section. Based on p-values from Figure 3, it seems pretty clear that MS is causal to UC. Also what is the difference between the CAUSE p-value of 0.16 in the text, and 6.97×10^{-7} in the Figure? Also (minor), I would be curious to know what are the LCV results (O'Connor et al. 2018 Nat Genet)
- p6 and Figure 4: It is hard to interpret the tissue results, as all the traits are significantly enriched for all the tissues, but the top tissue is same for MS and UC (spleen), and IBD and CD (lung). As I am pretty sure that all these gene-sets overlap by a lot, maybe analyses considering jointly baseline model + lung + spleen + all genes, or baseline model + lung + intestine + spleen + blood + all genes would help. It would be pretty interesting to show that once conditioned to each other, spleen is only significant for MS and UC, and lung for IBD and CD.
- p6 and Figure 5. Now the best cell-types are lung regulatory T cells for MS, and lung CD4+ T cells for MS. On figure 5, it is hard to conclude if there are multiple causal cell-types and/or shared causal cell-types, or if the high number of significant cell-types is due to correlation between the annotations. Maybe analyses considering jointly baseline model + lung regulatory T cells + lung CD4+ T cells + all genes, or baseline model + lung + spleen + lung regulatory T cells + lung CD4+ T cells + all genes would help.
- p6: could the authors mention in the text the datasets that have been used (i.e. GTEx for tissue analyses)?

- p16-l14: Did the authors also adjust to an annotation containing all the genes, as in Finucane et al. 2018 Nat Genet? This need to be done to correct for the fact that genes are important.

Reviewer #3 (Remarks to the Author):

I would first like to congratulate the authors with a clinically relevant question, and an extensive analysis. I think publication will help further research in the fields of complex disease genetics and chronic inflammatory diseases, while it can be of specific interest for IBD and MS researchers.

Major comments:

Pg 3 l13 – pg 5 l15:

Please weigh results to the contribution of the MHC region has made to them. The MHC region is highly variable genetically, a variation that is in no proportion to the variation in the rest of the genome. Functionally the relative impact of this variation is badly understood. Hence the regular practice in interpreting heritable impact is to report the MHC effect separately from that of the rest of the genomic variation.

Discussion:

Please bring the discussion back to the clinical importance of this research: the effect of MS treatment on the IBD phenotype and vice versa. Do results give any lead on the potential cause *(for examples associations with variants with opposite effect?)

Minor comments:

Many analyses have been performed and the reader would benefit from an extra sentence at each paragraph introducing the reasoning behind the analysis (in some paragraphs this is present, in others the methods and results are presented outright)

Pg 4 l 30-33:

What does this sentence mean exactly? “No evidence for a difference in (...) correlation” is a very uninformative formulation.

Pg 5 l 5-15:

I think “novel” loci does not help the reader in understanding the results, since I think you describe all overlapping SNPs here?

In addition: Are these the overlapping SNPs or are these the overlapping loci? Please specify and if these are overlapping SNPs rather than loci, please check whether these variants are indeed independent.

Pg 5 l 34- Pg 6 l 1-9:

Please provide an overview of the reference tissue available since the results are highly dependent on this.

Pg 6 l 11-28:

Very narrow reference of tissues, which makes these results hard to interpret in the wider scope of disease biology.

Pg 6 l 30-35:

“SMR applied to (..) identified 210 genome-wide significant associations”. Please split this sentence into two and spend part of the first on a more clear description of what the analysis actually entails, what it is meant to analyze.

Pg 8 ln 10-18:

Please clarify your description from “novel” to whether you describe all shared associations. If you are only describing the novel ones the unexpected shift in shared associations might be due to the associations already known.

Please also describe directionality of the shared associations.

As stated above, please assess independency of the associations.

Pg 8 ln 11-13:

You describe different directions, but are talking about 3 phenotypes with an association. Please describe which phenotype showed which direction and which were more consistent with each other.

Pg 8 ln 16-18:

Calculate whether this is significantly more in MS-CD than MS-UC than expected by evidence.

Pg 8 l 18:

Please clarify this statement: Why would the CD-MS correlation be more complex than the UC-MS correlation? Is it not more likely that the correlation is similar outside of the MHC region?

Reviewer #4 (Remarks to the Author):

This interesting study investigated the genetic relationships between MS, UC and CD through various approaches. They replicated genetic correlations estimates, performed cross-trait meta-analysis of GWAS, applied MR approaches to test potential causal relationships, and investigated the overlap in tissue- and cell-specific enrichment of SNP heritability. The main findings are updated genetic correlation estimates; 42 genome-wide significant SNPs in the cross-trait meta-analysis of MS and UC, CD or IBD; tissue enrichment in the small intestine for MS and UC; and various cell-type-specific enrichments consistent with the known biology of the diseases. The MR estimates are inconsistent and likely unreliable. The SMR results for MS were recently reported using the same dataset (PMID: 33005893).

I have the following comments for the authors to consider:

The genetic correlation between MS and IBDs is relatively modest compared to other traits to which MTAG has been applied (e.g. $r_g=0.7$ for trait pairs in the original MTAG paper). The reliability of this approach is unclear to me in this specific context, given r_g of 0.16 for MS-CD. This is especially true given that the 42 novel SNPs reported are not replicated. Given that the authors used the discovery cohort of the IMSSC Science 2019 publication for their MS analyses, is there an opportunity to use other MS datasets (ImmunoChip, MSChip) for replication of those SNPs? For example, the SNP rs13428812 reported in the MS-IBD and MS-CD analyses shows no association with MS susceptibility in the ImmunoChip (OR=1.027, $p=0.138$).

I am somewhat confused by the presentation of the MTAG results. The authors report that the SNPs identified are "novel loci shared between MS and IBD, UC and CD". However, my understanding is that the MTAG results are trait specific, and broadly equivalent to an increase in power of the single trait GWAS. In other words, the loci reported should be MS associated loci (if assumptions hold), not necessarily shared loci between MS and IBDs. Evidence of shared effects at a given locus would require a different approach than MTAG (e.g. colocalization). This confusion between trait-specific associations and 'shared effects' persists in the SMR results.

It is worth considering if MR is at all suited to this context. The no pleiotropy assumption is often a

strong and arguably unlikely assumption, but this seems especially true in the present context due to the genetic correlation between the 'exposure' and 'outcome'. While the correlation is not that strong as mentioned previously, it is thought to be strong enough for cross-trait meta-analysis. It is difficult to imagine that this correlation is exempt of pleiotropy. While some of the MR methods employed do account for pleiotropic effects under certain scenarios, the inconsistent results across methods likely suggests that most if not all are unable to account for these effects. The role of pleiotropy as a major limitation in this study is understated in the discussion. The authors state that "these effects [I assume pleiotropic effects] are likely negligible as we applied multiple MR approaches to minimise the false-positive rate of our results". I would argue that on the contrary, the often inconsistent results between the different MR methods are likely evidence of major pleiotropic effects, as discussed above.

Also, I am not sure I understand their example of pleiotropy through history of medication. This needs clarification.

How do the SMR results for MS compare to the recently published study by Jacobs et al. (PMID: 33005893)?

Minor comments:

In the introduction, the authors state the IFN-beta can increase the severity of IBD symptoms. However, studies of IFN treatment in IBD are conflicting and there is some evidence that it may improve symptoms in some patients (e.g. PMID: 12912859; reviewed in PMID: 32127733).

"A dilemma that doctors face in immunology and gastroenterology clinics is how to treat patients with both MS and IBD". Did the authors mean neuroimmunology instead of immunology? Neurologists often face this dilemma, more commonly than immunologists I would think.

Typo in introduction "Summary-date-based Mendelian randomization" instead of summary data-based Mendelian randomization.

I am not an expert in IBD, but I was surprised to see that the assumed prevalence of IBD was significantly lower than the sum of assumed prevalence of UC and CD. The reference cited (PMID: 27856364) suggests that the IBD prevalence should be closer to the sum of both (based on number of cases in Olmsted County).

We thank each of the Reviewer's for taking the time to review our manuscript, and for their helpful and constructive feedback which has greatly improved our paper. We have endeavoured to carefully and thoroughly address each of their comments and suggestions, as detailed in our response below.

Reviewer #2:

- In this paper, Yang et al. investigate the shared genetic architecture between multiple sclerosis (MS), and inflammatory bowel diseases (IBD), ulcerative colitis (UC), and Chron's disease (CD). They use state of the arts methods such as LD score regression, HESS, MTAG, numerous mendelian randomization methods, and SMR. I have two general comments: 1) While the paper is really easy to read for statistical geneticists, I believe that this paper would be harder to read for geneticists or physicians interested in the shared architecture of MS, IBD, UC and CD.

We thank the Reviewer and acknowledge that aspects of our original submission may have been inaccessible to readers outside the field of statistical genetics, including clinicians. We have carefully updated our manuscript to improve accessibility, including in relation to the motivation for the study, the rationale for each analysis, descriptions of the underlying theory for each statistical method, and interpretation of results. We have additionally included a new paragraph in the Discussion drafted by physician and joint-corresponding author Prof. Bruce Taylor to articulate the clinical importance of our results (page 12, line 14-34). While some statistical formulas and principles remain that may prove challenging for geneticists or physicians, these do not impact on the understanding of our article. We trust that our revised manuscript is now more accessible to a broader audience.

- 2) It is hard to understand what are the main conclusions. The abstract states that genetic correlation between MS and UC is twice the one between MS and CD, but this difference does not look significant (Figure 1), while mendelian randomization are pretty neat. Another example is that S-LDSC highlight many shared tissue and cell-type enrichments, but it is not clear if it is due to shared architecture, or high correlation between the corresponding annotations; similarly, only one gene is found shared using SMR.

We thank the Reviewer for raising this point, which has helped us to more clearly articulate our findings. We have now re-drafted our conclusions in the Abstract (page 2) and Discussion (page 13, line 14-24) to summarise four key results from our paper:

1. Identification of a significantly higher genetic correlation between MS and UC compared to that between MS and CD;
2. Identification of 9 novel SNPs shared between MS and IBD (N=5; of which two were also significant for MS-CD), UC (N=2), and CD (N=4) and 3 genes (i.e. *GPR25*, *AL031282.2* and *AL109917.1*) shared between MS and IBD or UC, using cross-trait GWAS meta-analysis and SMR (respectively);
3. Identification of a suggestive (but inconclusive) causal effect of MS on IBD and UC, but not CD;
4. Consistent observation of specific tissues and cell types showing SNP heritability enrichments for MS and IBDs, which were largely robust to conditioning on other tissues and cell types.

- Here are several presentation and methodological suggestions that could correct these two points: (minor) p4-114: Numerous papers shown that LDSC under-estimates the h^2 of a trait (Speed et al. 2017 Nat Genet, Gazal et al. 2019 Nat Genet, Hou et al. 2019 Nat Genet), but this can be corrected using S-LDSC with the baseline-LD model (Gazal et al. 2019 Nat Genet, Speed et al. 2020 Nat Genet).

We thank the Reviewer for this suggestion, with which we agree. We have updated our SNP heritability estimates for MS, IBD, UC and CD using S-LDSC with the baseline-LD model (page 4, line 19-21). As expected, all SNP heritability estimates were slightly higher than in our original submission: specifically, h^2_{SNP} increased from 13% to 16% for MS, from 16% to 19% for IBD, from 15% to 17% for UC and from 25% to 26% for CD. We also added additional text to the Methods to provide a brief description of the baseline-LDSC method (page 15 line 14-20).

- p4-119: what is the statistical significance between the genetic correlation between MS and UC and the one between MS and CD? Figure 1 suggests that it is not significant.

We thank the Referee for noting this omission. We have now calculated the significance of the difference of the genetic correlations between MS and UC and MS and CD. Specifically, we first calculated Fisher's transformed Z-score from r_g using the formula $Z\text{-score} = \frac{Z_{r_{g1}} - Z_{r_{g2}}}{s}$, where r_{g1} and r_{g2} represents the two genetic correlations, Fisher's transformed $Z_{r_g} = \frac{1}{2} \ln\left(\frac{1+r_g}{1-r_g}\right)$ for each correlation, $s = \sqrt{\frac{1}{n_{r_{g1}}-3} + \frac{1}{n_{r_{g2}}-3}}$, with the effective sample size per correlation given by $n_{r_g} = \frac{1-r_g}{s \cdot e_{r_g}^2} + 2$. We then calculated the two-tailed p-value from the Z-score of a standard normal distribution. Using this approach, we found that the genetic correlation between MS and UC was significantly greater than that between MS and CD, both with (Z-score=1.97, p=0.05) and without (Z-score=2.39, p=0.02) the intercept constrained. We have added these results into the main text (page 4, line 24-25) and Table S1, and we have described the approach for calculating the significance of the difference in the Methods (page 15 [line 27] to page 16 [line 2]).

- p4-118: not clear to the reader what is the difference between LDSC with intercept and without intercept.

Thank you for bringing this to our attention. We have clarified the difference between unconstrained intercept LDSC and constrained intercept LDSC in light of the reviewer's comment, both in the Methods (page 16, line 4-9) and Results (page 4, line 29-31). The relevant revised Methods section now reads:

“As a sensitivity analysis, we also performed LDSC with the single-trait heritability intercept constrained. Compared to LDSC with unconstrained intercept, constrained intercept LDSC is an approach designed to decrease the standard error of estimates under the assumption of no population stratification, thereby indirectly evaluating the influence of GWAS statistic inflation. However, constrained intercept LDSC may also provide a biased or misleading estimate of heritability and genetic correlation if the intercept is constrained incorrectly.”

- p4-129: not clear to the reader what is the difference between genetic correlation and local genetic correlation.

We apologise for failing to make this distinction clearer in our original submission. We have clarified the definition of ‘genetic correlation’ and ‘local genetic correlation’ in the LDSC and ρ -HESS sections of the Methods (page 15, line 27-29 & page 16, line 13-15). The revised sections now read:

“We used bivariate LDSC to estimate genetic correlations (r_g , i.e. the proportion of genetic variance shared by two traits divided by the square root of the product of their SNP heritability estimates) between MS and each of IBD, UC and CD, as well as between UC and CD” and “... we applied ρ -HESS to evaluate the local genetic correlations (i.e. genetic correlation between traits due to their shared genetic variance in a defined genomic region) between MS and each of IBD, UC, and CD.”

We have also added the definition of SNP heritability (page 15, line 14-15; “ h^2_{SNP} , i.e. the proportion of phenotypic variance in a trait can be explained by common genetic variants tagged on SNP arrays.”).

- p5-117: not sure to get the title of this section. Based on p-values from Figure 3, it seems pretty clear that MS is causal to UC. Also what is the difference between the CAUSE p-value of 0.16 in the text, and 6.97×10^{-7} in the Figure? Also (minor), I would be curious to know what are the LCV results (O'Connor et al. 2018 Nat Genet).

We apologise if this section of the manuscript lacked clarity, and in particular for any confusion in relation to the different p-values reported from the CAUSE analysis. CAUSE fits three models for each pair of traits (i.e. MS-IBD, MS-UC, MS-CD); namely, a causal model (i.e. instrumental SNPs act on exposure and outcome through a causal pathway and shared factors), a sharing model (i.e. instrumental SNPs act on exposure and outcome only through shared factors), and a null model (i.e. no causal pathway or shared factors underlying exposure and outcome). Here, the p-value of 0.16 was calculated from the test comparing the model fitness of the causal model and sharing model for MS-UC. This suggested that the causal model was not superior to the sharing model (at a 5% significance level) and thus that CAUSE was unable to distinguish causality from correlated pleiotropy. These results were inconsistent with those from the other five MR methods, thus we conservatively stated that we find ‘suggestive but inconclusive evidence for causality between MS and UC’. The p-value of 6.97×10^{-7} was the p-value of the CAUSE causal model for MS-UC (displayed in Figure 3).

In response to the Reviewer’s query about LCV, we applied this method to MS-IBD, MS-UC, and MS-CD, finding that the LCV genetic causality proportion (GCP) was non-significant in each case (see table below). We chose not to include LCV in our original MR analysis, for two reasons: first, LCV can lead to conservative p-values when the genetic correlation is low, and this is the case between MS and each of IBD, UC, and CD (i.e. $r_g \sim 0.10$ to 0.30). Second, LCV assumes that the genetic correlation between two traits is mediated by a latent variable having a causal effect on each trait, and the method quantifies this latent causal variable using the genetic causality proportion.

This is similar to the method implemented in CAUSE, but whereas CAUSE provides a formal test for causality, this is not the case for LCV.

Trait1 – Trait2	GCP (SE)	GCP p-value	RHO (SE)	p-value (fully causal)
MS – IBD	-0.01 (0.14)	0.96	0.20 (0.06)	7.99×10^{-38} , 1.04×10^{-36}
MS – UC	-0.01 (0.21)	0.89	0.30 (0.04)	8.84×10^{-27} , 1.17×10^{-14}
MS – CD	-0.45 (0.33)	0.44	0.05 (0.08)	1.14×10^{-123} , 0.39

- p6 and Figure 4: It is hard to interpret the tissue results, as all the traits are significantly enriched for all the tissues, but the top tissue is same for MS and UC (spleen), and IBD and CD (lung). As I am pretty sure that all these gene-sets overlap by a lot, maybe analyses considering jointly baseline model + lung + spleen + all genes, or baseline model + lung + intestine + spleen + blood + all genes would help. It would be pretty interesting to show that once conditioned to each other, spleen is only significant for MS and UC, and lung for IBD and CD.

- p6 and Figure 5. Now the best cell-types are lung regulatory T cells for MS, and lung CD4+ T cells for MS. On figure 5, it is hard to conclude if there are multiple causal cell-types and/or shared causal cell-types, or if the high number of significant cell-types is due to correlation between the annotations. Maybe analyses considering jointly baseline model + lung regulatory T cells + lung CD4+ T cells + all genes, or baseline model + lung + spleen + lung regulatory T cells + lung CD4+ T cells + all genes would help.

We are grateful to the Reviewer for this insightful suggestion. We agree that performing conditional S-LDSC analyses is important in order to establish if the specific tissues and cell types that we report as FDR-significant are individually associated with disease. We conducted conditional S-LDSC analyses for each FDR-significant GTEx tissue and cell type reported in our original submission. In the case of tissue-level analyses, we conditioned on the baseline model, the set of all genes and the set of genes highly expressed in other FDR-significant tissues. For example, in conditional S-LDSC analyses for lung, we adjusted for the baseline model, the set of all genes and the set of genes highly expressed in small intestine - terminal ileum, spleen and whole blood. In the case of cell type-specific conditional S-LDSC analyses, we additionally conditioned on the set of genes highly expressed in the other FDR-significant cell types of the same tissue, for the focal disease. For example, in the case of B hypermutation cells (spleen) in MS, conditional S-LDSC adjusted for the baseline model, the set of all genes, the set of genes highly expressed in small intestine - terminal ileum, lung and whole blood, and the set of genes highly expressed in B/T doublet cells (spleen) and CD8⁺ cytotoxic lymphocytes (spleen).

In the tissue-level conditional S-LDSC analyses, the heritability enrichments in all four tissues remained FDR- ($p < \sim 5 \times 10^{-3}$) or Bonferroni- ($p < \sim 3 \times 10^{-4}$) significant for MS and IBD. Interestingly, the only FDR-significant tissue for CD in the conditional analyses was lung, whereas for UC we observed FDR-significant enrichment in spleen and small intestine-terminal ileum. In the cell type-specific conditional S-LDSC analyses, CD8⁺ cytotoxic T cells in lung and spleen showed FDR-significant heritability enrichment for both MS and CD. Another six cell types also exhibited FDR-significant heritability enrichment, but for only one of MS, IBD or UC, including MS-specific

enrichment in dividing T cells in lung and enterocyte progenitors in small intestine, IBD-specific enrichment in activated dendritic cells in lung and CD56⁺ NK cells in peripheral blood, and UC-specific enrichment in activated dendritic cells in lung and early enterocytes in small intestine. These results add further novelty to our manuscript.

We also performed a parallel set of *stricter* conditional S-LDSC analyses that adjusted tissue-level (or cell type-specific level) heritability enrichment by the baseline model, the set of all genes and the set of genes highly expressed in all other GTEx tissues (or individual cell types), other than the focal tissue (or cell type). Tissue (and cell-type) heritability enrichments in these analyses were by-and-large very similar to the initial set of conditional S-LDSC analyses.

Finally, we performed a series of conditional gene-set enrichment analyses in MAGMA (see Supplementary Notes), and observed very similar results, suggesting that our conditional S-LDSC results are robust.

We have updated the relevant sections of the Methods (page 19 [line 26] to page 20 [line 3]), Results (page 6, line 22-31; page 7, line 19-27) and Discussion (page 9, line 30-33; page 10, line 19-20 & 31-34; page 11, line 7-16), together with Figures 4 and 5, Supplementary Figures S11, S12, S35, S36, S38 & S39 and Tables S12, S13, S16 & S17, to incorporate these changes.

- p6: could the authors mention in the text the datasets that have been used (i.e. GTEx for tissue analyses)?

Details of all datasets used in our analyses, included GTEx and multiple single cell RNA-seq datasets, are provided in the Methods (page 14 [line 26] to page 15 [line 10]).

- p16-114: Did the authors also adjust to an annotation containing all the genes, as in Finucane et al. 2018 Nat Genet? This need to be done to correct for the fact that genes are important.

Yes, all of our S-LDSC analyses were adjusted for the set of all genes described in Finucane et al. 2018, as described in the main text (page 19, line 16-19).

Reviewer #3:

- I would first like to congratulate the authors with a clinically relevant question, and an extensive analysis. I think publication will help further research in the fields of complex disease genetics and chronic inflammatory diseases, while it can be of specific interest for IBD and MS researchers.

We thank the Reviewer for acknowledging the relevance of our contribution to the field.

- Major comments: Pg 3 l13 – pg 5 l15:

Please weigh results to the contribution of the MHC region has made to them. The MHC region is highly variable genetically, a variation that is in no proportion to the variation in the rest of the genome. Functionally the relative impact of this variation is badly understood. Hence the regular practice in interpreting heritable impact is to report the MHC effect separately from that of the rest of the genomic variation.

We thank the Reviewer and agree that genetic variation in the MHC region is important to liability for MS and IBDs. However, we note that some genetic analyses of the MHC can be challenging using currently available statistical methods, due to the complex and long-range LD in this region. For example, LDSC, bivariate LDSC (for estimation of genetic correlation) and S-LDSC (for estimation of tissue- and cell type-specific SNP heritability enrichment) are all sensitive to the complicated LD in the MHC, and it is recommended (by the developers) that the MHC be excluded from these analyses. This is also the case for SMR. For these reasons, many (but not all) of the analyses in our original submission either excluded the MHC region or included the MHC region as a form of sensitivity analysis (e.g. MR analyses with and without MHC SNPs). Exceptions include MTAG and ρ -HESS, which we implemented with MHC SNPs included. We note that the developers of MTAG¹ and ρ -HESS² do not explicitly refer to the MHC region, but some studies³ implementing these methods chose to exclude MHC SNPs when performing ρ -HESS.

In response to the Reviewer's comment, our updated manuscript now includes additional SMR results including MHC SNPs, with the caveat that many fail the HEIDI test, and with the outcome that this approach failed to identify any MHC genes associated with both MS and IBD, UC or CD (page 20, line 22-26; page 8, line 12-14). We also implemented MAGMA for tissue- and cell type-specific enrichment analyses including the MHC (Methods and Results are described in the Supplementary Notes). These analyses revealed slightly higher heritability enrichments with the inclusion of genes in the MHC region (Table S14-15), suggesting a moderate contribution of the MHC region to shared liability of MS and IBDs. Collectively, our analyses are consistent with a moderate contribution of genetic variation in the MHC region to the shared genetics underlying MS and IBDs. Nevertheless, quantifying the genetic contribution of the MHC region in disease is challenging and further studies are warranted. We have updated the Discussion to reflect these points (page 12, line 4-12).

*- Discussion: Please bring the discussion back to the clinical importance of this research: the effect of MS treatment on the IBD phenotype and vice versa. Do results give any lead on the potential cause *(for examples associations with variants with opposite effect?).*

We thank the reviewer for this suggestion. We have added a paragraph in the Discussion to emphasize the clinical implications of our research (page 12, line 14-34):

“The clinical implications of our findings are multifaceted. First, they help to explain the differential effects of immunomodulating therapies between MS and IBD. Previously researchers have found that type I interferons (including interferon- β , a pivotal MS therapy until recently) have tissue-specific and gene-specific effects that are dependent on a balance of different cytokines. Although the immunomodulatory mechanism for interferon- β therapy in MS is not fully understood, its key role is in presumably inhibiting transmigration of autoreactive T cells into the CNS. Given its tissue-specific effects, interferon- β treatment may unexpectedly lead to immunopathology in other tissues by influencing dendritic cell activation and maturation, and natural killer cell activity; both of these cell types showed significant heritability enrichments for IBD in our study, potentially explaining the deleterious effect of these medications in some cases of IBD. Similarly, anti-TNF therapy for IBD can be associated with the development of MS like lesions in the CNS and/or worsening of extant MS, although the mechanism is unclear. Anti-TNF therapy does reduce CD8⁺ T cell proliferation, which can result in viral infection or reactivation. In our study, CD8⁺ cytotoxic T cells in lung showed significant heritability enrichment for MS while it is well known that elevated EBNA-1 IgG level (due to Epstein-Bar virus infection or reactivation) is an important risk factor for MS development and progression. Therefore, fine tuning current treatments or developing new therapies that are capable of delivering targeted effects in disease-specific tissues and/or cell types may be a solution to tackle the problem of concomitant worsening of co-morbid MS or IBD with some therapies. What is more, our findings can be used as a foundation for future studies to map causal tissues, cells and genes to gain new insights into the pathogenesis of these diseases, which ultimately could lead to novel drug treatments that may improve clinical outcomes.”

In relation to your question about genetic directionality, please refer to our response to your related question (*Pg 8 ln 11-13*) below. In brief, all the novel cross-trait SNPs have the same direction of effect in the component GWAS's. There was a misinterpretation of this analysis in our original submission, which has been corrected in this revision.

- Minor comments: Many analyses have been performed and the reader would benefit from an extra sentence at each paragraph introducing the reasoning behind the analysis (in some paragraphs this is present, in others the methods and results are presented outright).

We thank the Reviewer for this suggestion. We have added an extra sentence at the beginning of each Methods and Results section to describe the rationale for the analysis. For example, the revised Results section for the SMR analysis (page 7, line 30-32) now reads:

“We applied SMR to identify putative functional genes underlying MS and each of IBD, UC and CD, by jointly analysing GWAS summary data for MS, IBD, UC and CD and eQTL summary data from eQTLGen (whole blood) and GTEx (lung, small intestine-terminal ileum, spleen).”

- Pg 4 l 30-33: What does this sentence mean exactly? “No evidence for a difference in (...) correlation” is a very uninformative formulation.

We apologise if this was not clear. We have re-worded this sentence to improve clarity (page 4 [line 35] to page 5 [line 3]), as follows:

“In each of the three pairwise comparisons (MS-IBD, MS-UC, MS-CD), there was no evidence for a difference in the average local genetic correlation in regions harbouring MS-specific loci versus IBD-, UC- and CD-specific loci (Figure 2)”

- Pg 5 l 5-15: I think “novel” loci does not help the reader in understanding the results, since I think you describe all overlapping SNPs here? In addition: Are these the overlapping SNPs or are these the overlapping loci? Please specify and if these are overlapping SNPs rather than loci, please check whether these variants are indeed independent.

We are grateful to the Reviewer for this insightful comment. In our original submission, we defined novel SNPs in the cross-trait GWAS meta-analyses (e.g. MS-IBD) as the set of LD-pruned ($LD\ r^2 < 0.05$ within 1,000-kb windows, based on the UK Biobank European reference imputed to the Haplotype Reference Consortium) genome-wide significant ($p < 5 \times 10^{-8}$) SNPs that were not genome-wide significant in the contributing single-trait GWAS (e.g. MS [discovery cohort] or IBD). As a consequence of the Reviewer’s perceptive comment, we realised that we had inadvertently overlooked the possibility that some of these SNPs may be in LD with genome-wide significant SNPs reported in the single-trait GWAS.

In our revised manuscript, novel SNPs are now defined using the two criteria described above, together with another filter to exclude SNPs in LD (i.e. $LD\ r^2 > 0.05$ within 1,000-kb windows using the same reference) with any genome-wide significant SNP in the original GWAS of MS (discovery cohort [14,802 cases, 26,703 controls] in the IMSGC MS GWAS⁴), IBD⁵, UC⁵, and CD⁵, as well as the N=200 non-MHC genome-wide significant SNPs reported from the full IMSGC GWAS meta-analysis of MS (discovery + replicate cohorts [47,429 cases, 68,374 controls]⁴), for which we do not have access to the complete GWAS summary statistics.

In our updated analysis (page 5, line 12-26), we identified 9 novel SNPs for MS-IBD (N=5; of which two were also significant for MS-CD), MS-UC (N=2) and MS-CD (N=4). We have updated the relevant sections of the Methods (page 16 [line 25] to page 17 [line 8]), Results (page 5, line 12-26) and Discussion (page 9, line 8-13), together with supplementary Tables S3, to incorporate these changes. Once again, we thank the Reviewer for bringing our attention to this issue.

- Pg 5 l 34- Pg 6 l 1-9: Please provide an overview of the reference tissue available since the results are highly dependent on this.

We have now updated the Methods section describing the GTEx data (page 14, line 27-30).

- Pg 6 l 11-28: Very narrow reference of tissues, which makes these results hard to interpret in the wider scope of disease biology.

We thank the Reviewer for this comment. To our knowledge, GTEx remains the largest publicly available resource of tissue-level gene expression data, with 53 tissues collected from approximately 700 non-diseased human donors. As noted in our response to Reviewer #2 (*p6 and Figure 4*), we performed conditional S-LDSC to correct for potential gene overlap among tissues

and found that heritability enrichment for lung, spleen, small intestine and whole blood in MS and IBDs remained significant. These analyses (see details above) strengthen the evidence in our manuscript for SNP heritability enrichments specifically in immune system-related tissues.

- Pg 6 l 30-35: “SMR applied to (..) identified 210 genome-wide significant associations”. Please split this sentence into two and spend part of the first on a more clear description of what the analysis actually entails, what it is meant to analyze.

We apologise if this was not clear in the manuscript. We modified the sentence (page 7, line 30-35) in light of the Reviewer’s comment, which now reads:

“We applied SMR to identify putative functional genes underlying MS and each of IBD, UC and CD, by jointly analysing GWAS summary data for MS, IBD, UC and CD and eQTL summary data from eQTLGen (whole blood) and GTEx (lung, small intestine–terminal ileum, spleen). We identified 210 non-MHC genome-wide significant associations ($p_{\text{SMR}} < 5.32 \times 10^{-7}$), of which 59 (representing 41 unique genes) survived the HEIDI (HEterogeneity In Dependent Instrument)-outlier test (Table S19, Figures S40-43).”

- Pg 8 ln 10-18: Please clarify your description from “novel” to whether you describe all shared associations. If you are only describing the novel ones the unexpected shift in shared associations might be due to the associations already known. Please also describe directionality of the shared associations. As stated above, please assess independency of the associations.

We thank the Reviewer for this remark, which was also raised by Reviewer #4, and refer to our response to your previous comment (*Pg 5 l 5-15*), in which we clarify (and correct) the definition of novel SNPs, including in relation to their independence. Please see our response to your next comment for details about the directionality of these shared associations.

- Pg 8 ln 11-13: You describe different directions, but are talking about 3 phenotypes with an association. Please describe which phenotype showed which direction and which were more consistent with each other.

We thank the Reviewer for noting the need for further clarification on this point. As noted in our response to an earlier comment from the Reviewer (*Pg 5 l 5-15*) we have revised the definition of novel SNPs in the cross-trait GWAS meta-analyses (see above). We have additionally clarified the effect direction of the 9 novel SNPs identified in these MTAG analyses (page 9, line 9-13). Our revised text now reads:

“Notably, all 9 SNPs showed consistent direction of effect between the cross-trait GWAS meta-analyses and the component single trait GWAS’s (Table S3). Moreover, all these SNPs, including the two novel SNPs shared by MS-IBD and MS-CD (i.e. rs13428812 and rs4944014), also showed the same direction of effect in all three cross-trait meta-analyses (Table S3). The results suggest that these novel SNPs are likely involved in regulating common pathways shared between MS and IBDs.”

- Pg 8 ln 16-18: Calculate whether this is significantly more in MS-CD than MS-UC than expected by evidence.

We thank the Reviewer for this suggestion, which was also made by Reviewer #2. We have performed a significance test to compare the genetic correlations between MS and UC and MS and CD. The genetic correlation between MS and UC was found to be significantly greater than that between MS and CD, irrespective of whether the LDSC intercept was constrained or not (without constrained intercept: Z-score=2.39, p=0.02; constrained intercept: Z-score=1.97, p=0.05). We have added these results (page 4, line 24-25) and methods (page 15 [line 27] to page 16 [line 2]) to the main text and Table S1.

- Pg 8 l 18: Please clarify this statement: Why would the CD-MS correlation be more complex than the UC-MS correlation? Is it not more likely that the correlation is similar outside of the MHC region?

We thank the Reviewer for this comment, with which we agree. We have now revised the paragraph describing the novel SNPs identified by MTAG and this statement has been removed.

Reviewer #4:

- This interesting study investigated the genetic relationships between MS, UC and CD through various approaches. They replicated genetic correlations estimates, performed cross-trait meta-analysis of GWAS, applied MR approaches to test potential causal relationships, and investigated the overlap in tissue- and cell-specific enrichment of SNP heritability. The main findings are updated genetic correlation estimates; 42 genome-wide significant SNPs in the cross-trait meta-analysis of MS and UC, CD or IBD; tissue enrichment in the small intestine for MS and UC; and various cell-type-specific enrichments consistent with the known biology of the diseases. The MR estimates are inconsistent and likely unreliable. The SMR results for MS were recently reported using the same dataset (PMID: 33005893).

We thank the Reviewer for taking the time to carefully review our manuscript.

- I have the following comments for the authors to consider:

The genetic correlation between MS and IBDs is relatively modest compared to other traits to which MTAG has been applied (e.g. $rg=0.7$ for trait pairs in the original MTAG paper). The reliability of this approach is unclear to me in this specific context, given rg of 0.16 for MS-CD. This is especially true given that the 42 novel SNPs reported are not replicated. Given that the authors used the discovery cohort of the IMSGC Science 2019 publication for their MS analyses, is there an opportunity to use other MS datasets (ImmunoChip, MSChip) for replication of those SNPs? For example, the SNP rs13428812 reported in the MS-IBD and MS-CD analyses shows no association with MS susceptibility in the ImmunoChip ($OR=1.027$, $p=0.138$).

We thank the Reviewer for these insightful comments and apologise if the MTAG methods were unclear in our original submission. In our study, we used MTAG options that implemented a method equivalent to inverse-variance weighted meta-analyses (i.e. implicitly assuming equal SNP heritability for each trait and perfect genetic covariance between traits). Details of these MTAG options can be found at <https://github.com/JonJala/mtag/wiki/Tutorial-2:-Special-Options>. We chose to implement this version of MTAG, rather than conventional inverse-variance weighted meta-analyses (e.g. in METAL), because MTAG can accommodate potential sample overlap between traits estimated by LDSC (page 16, line 26-30). This approach has been applied by other studies^{6,7}. We chose not to perform trait-specific MTAG because of the moderate genetic correlations between MS and IBDs (as noted by the Reviewer).

As detailed in our response to a perceptive comment from Reviewer #3 (Pg 5 l 5-15), we realised that we had inadvertently omitted a filtering step in the definition of novel SNPs from the cross-trait GWAS meta-analyses. Whereas originally we defined novel SNPs as the set of LD-pruned genome-wide significant ($p < 5 \times 10^{-8}$) SNPs that were not genome-wide significant in the contributing single-trait GWAS, in our revised manuscript we additionally filter any SNP in LD with any genome-wide significant SNP in the original GWAS of MS (discovery cohort [14,802 cases, 26,703 controls] in the IMSGC MS GWAS⁴), IBD⁵, UC⁵, and CD⁵, as well as the N=200 non-MHC genome-wide significant SNPs reported from the full IMSGC GWAS meta-analysis of MS (discovery + replicate cohorts [47,429 cases, 68,374 controls]⁴), for which we do not have access to the complete GWAS summary statistics. In this revised analysis, we identified 9 novel SNPs for MS-IBD (N=5; of which two were also significant for MS-CD), MS-UC (N=2), and MS-CD (N=4). We have updated the relevant sections of the Methods (page 16 [line 25] to page 17 [line 8]), Results (page 5, line 12-26)

and Discussion (page 9, line 8-13), together with supplementary Tables S3, to incorporate these changes.

Finally, we thank the Reviewer for the suggestion of using other MS datasets (e.g. ImmunoChip, MSChip) for replication. However, because our MTAG analyses are equivalent to standard cross-trait meta-analysis, the novel SNPs identified are associated with the cross-trait phenotypes (i.e. MS-IBD, MS-UC, or MS-CD), as opposed to individual diseases (e.g. MS). For this reason, the use of independent MS-specific GWAS datasets would not be equivalent to replication. The latter would require access to large independent GWAS summary data for MS and one or more of IBD, UC or CD, which to our knowledge is not currently available.

- I am somewhat confused by the presentation of the MTAG results. The authors report that the SNPs identified are "novel loci shared between MS and IBD, UC and CD". However, my understanding is that the MTAG results are trait specific, and broadly equivalent to an increase in power of the single trait GWAS. In other words, the loci reported should be MS associated loci (if assumptions hold), not necessarily shared loci between MS and IBDs. Evidence of shared effects at a given locus would require a different approach than MTAG (e.g. colocalization). This confusion between trait-specific associations and 'shared effects' persists in the SMR results.

We apologise if this section of the manuscript lacked clarity. As explained above, we used MTAG to perform the equivalent of inverse-variance weighted meta-analyses, which is similar to methods implemented in METAL, but with the important difference that MTAG can adjust for sample overlap between traits. We did not conduct trait-specific MTAG because the genetic correlations between MS and IBDs are significantly less than 0.7 (the threshold recommended by the MTAG paper) and thus probably underpowered for identification of novel trait-specific SNPs.

- It is worth considering if MR is at all suited to this context. The no pleiotropy assumption is often a strong and arguably unlikely assumption, but this seems especially true in the present context due to the genetic correlation between the 'exposure' and 'outcome'. While the correlation is not that strong as mentioned previously, it is thought to be strong enough for cross-trait meta-analysis. It is difficult to imagine that this correlation is exempt of pleiotropy. While some of the MR methods employed do account for pleiotropic effects under certain scenarios, the inconsistent results across methods likely suggests that most if not all are unable to account for these effects. The role of pleiotropy as a major limitation in this study is understated in the discussion. The authors state that "these effects [I assume pleiotropic effects] are likely negligible as we applied multiple MR approaches to minimise the false-positive rate of our results". I would argue that on the contrary, the often inconsistent results between the different MR methods are likely evidence of major pleiotropic effects, as discussed above.

We thank the Reviewer for these constructive comments. We agree that pleiotropy is an important limitation of our MR analyses, as is the case in many applications of these methods. It is precisely because of the difficulty of differentiating causality from pleiotropy that we conservatively based our inference on six distinct and highly-cited MR methods, each making different assumptions about pleiotropy, as opposed to only one or two, which we note is not uncommon in the field. We found consistent evidence for a causal effect of MS on UC and IBD using five of six methods, all of which surpassed Bonferroni significance. The one exception was the CAUSE method, which could

not distinguish a causal effect of MS on UC or IBD from correlated and/or uncorrelated pleiotropy. Taking into account all of these results, we adopted a conservative interpretation, which is that evidence for causality of MS in relation to UC is suggestive but inconclusive. At the time we felt this was a careful position that did not over-interpret the data. Having now received the Reviewer's comments, and having reflected on the fact that CAUSE is the only MR method capable of distinguishing causality from correlated pleiotropy, we acknowledge that we may have understated the importance of pleiotropy in the discussion. We have revised the relevant sections of the Discussion to reflect this position (page 9, line 17-20 & page 13, line 1-8), which now read:

Page 9 (line 17-20):

“A consistent causal effect of MS on UC (and IBD) was inferred using five of six MR methods with the exception of CAUSE. Since CAUSE is the only MR method capable of distinguishing causality from correlated pleiotropy, we could not further rule out the possibility of horizontal pleiotropy, which is a limitation of our analysis (see below).”

Page 13 (line 1-8):

“First, with reference to our MR analyses, we note that CAUSE is the only MR method (of the six implemented in our study) capable of distinguishing causality from correlated pleiotropy (i.e. a shared heritable factor affecting both exposure and outcome), and thus the latter may explain the inconsistent inference between CAUSE and the other MR methods in relation to a causal effect of MS on both UC and IBD. Evidence suggests that correlated pleiotropy is common and may frequently contribute to false positive inference in Mendelian randomisation. Further investigations in larger, more powerful datasets will be needed to determine if MS is causal for IBDs.”

- Also, I am not sure I understand their example of pleiotropy through history of medication. This needs clarification.

We used history of medication as an example of unmeasured correlated pleiotropy, because a recent paper⁸ noted that the causal relationship between systolic blood pressure and coronary heart disease may be biased by ibuprofen intake levels (i.e. because patients with severe headache tend to have higher ibuprofen intake levels that increase systolic blood pressure and thus also coronary heart disease). We have now acknowledged the possibility that correlated pleiotropy is responsible for the inconsistent inference between CAUSE and other MR methods in our analyses. However, we don't believe that the history of medication example adds anything to our discussion, and in fact, as noted by the Reviewer, may be confusing, since we do not intend to suggest that history of medication is specifically responsible for the difference in inference between CAUSE and other MR methods in our analyses of MS and IBDs. For this reason, we have removed the history of medication example from the revised manuscript.

- How do the SMR results for MS compare to the recently published study by Jacobs et al. (PMID: 33005893)?

We thank the Reviewer for bringing our attention to the recent paper by Jacobs *et al.*, which we were not aware of at the time of submission. Our SMR results, which are a relatively minor component of our manuscript, are consistent with their study, as we utilised the same MS GWAS

and eQTLGen summary data. We have now added additional text to the Discussion to acknowledge that our SMR findings are consistent with those of Jacobs *et al.* 2020 (page 11, line 29-31).

- *Minor comments: In the introduction, the authors state the IFN-beta can increase the severity of IBD symptoms. However, studies of IFN treatment in IBD are conflicting and there is some evidence that it may improve symptoms in some patients (e.g. PMID: 12912859; reviewed in PMID: 32127733).*

We thank the Reviewer for this comment. We have now revised the relevant text (page 3 [line 31] to page 4 [line 3]) to read:

“For example, it has been reported that the cytokine, interferon- β , used to treat MS can increase the severity of IBD symptoms (although findings are conflicting with some evidence suggesting that interferon treatment may improve symptoms in some IBD patients). Additionally, others have found that TNF- α antagonists that are effective for IBD can worsen the clinical course of MS. These important and not uncommon clinical questions may be helped by an improved understanding of genetic relationships between MS and comorbid IBD that would lead to safer and more effective interventions for both diseases individually and when they occur together.”

- *"A dilemma that doctors face in immunology and gastroenterology clinics is how to treat patients with both MS and IBD". Did the authors mean neuroimmunology instead of immunology? Neurologists often face this dilemma, more commonly than immunologists I would think.*

We thank the Reviewer for this suggestion. We have changed “immunology” to “neuroimmunology”.

- *Typo in introduction "Summary-date-based Mendelian randomization" instead of summary data-based Mendelian randomization.*

We thank the Reviewer for pointing out this typo, which we have corrected.

- *I am not an expert in IBD, but I was surprised to see that the assumed prevalence of IBD was significantly lower than the sum of assumed prevalence of UC and CD. The reference cited (PMID: 27856364) suggests that the IBD prevalence should be closer to the sum of both (based on number of cases in Olmsted County).*

We thank the Reviewer for identifying what was a typo in our original submission. We have now changed the population prevalence for IBD to 0.54%⁹, and all liability-scale results for IBD have been updated.

References

1. Turley, P. *et al.* Multi-trait analysis of genome-wide association summary statistics using MTAG. *Nat Genet* **50**, 229-237 (2018).
2. Shi, H., Mancuso, N., Spendlove, S. & Pasaniuc, B. Local Genetic Correlation Gives Insights into the Shared Genetic Architecture of Complex Traits. *Am J Hum Genet* **101**, 737-751 (2017).
3. Tylee, D.S. *et al.* Genetic correlations among psychiatric and immune-related phenotypes based on genome-wide association data. *Am J Med Genet B Neuropsychiatr Genet* **177**, 641-657 (2018).
4. International Multiple Sclerosis Genetics Consortium. Multiple sclerosis genomic map implicates peripheral immune cells and microglia in susceptibility. *Science* **365**(2019).
5. Liu, J.Z. *et al.* Association analyses identify 38 susceptibility loci for inflammatory bowel disease and highlight shared genetic risk across populations. *Nat Genet* **47**, 979-986 (2015).
6. van der Harst, P. & Verweij, N. Identification of 64 Novel Genetic Loci Provides an Expanded View on the Genetic Architecture of Coronary Artery Disease. *Circ Res* **122**, 433-443 (2018).
7. Han, X. *et al.* Genome-wide association analysis of 95 549 individuals identifies novel loci and genes influencing optic disc morphology. *Hum Mol Genet* **28**, 3680-3690 (2019).
8. Cho, Y. *et al.* Exploiting horizontal pleiotropy to search for causal pathways within a Mendelian randomization framework. *Nat Commun* **11**, 1010 (2020).
9. Shivashankar, R., Tremaine, W.J., Harmsen, W.S. & Loftus, E.V., Jr. Incidence and Prevalence of Crohn's Disease and Ulcerative Colitis in Olmsted County, Minnesota From 1970 Through 2010. *Clin Gastroenterol Hepatol* **15**, 857-863 (2017).

REVIEWER COMMENTS

Reviewer #2 (Remarks to the Author):

The authors made a great work at answering my concerns, especially the ones regarding paper clarity and S-LDSC analyses.

I am particularly impressed by the suggested conditional analyses, showing distinct tissues in CD and UC.

Really minor points:

page 4 -line 17: what do you mean by baseline ldsc. Do you mean mean stratified ldsc? (as it seems that you use the baseline-LD model to estimate heritability)

page 6 - line 28: whole blood looks also significant for CD in Table S12.

Reviewer #3 (Remarks to the Author):

--The changes that the authors made to the manuscript have greatly improved the readability, especially for non-bioinformaticians. All the issues I touched upon in my comments have been largely addressed to my satisfaction. I would like to comment to the authors on two of these (see below). -- In addition, I would like to suggest to the authors to maybe limit the number of results described in the abstract. I understand that the study does not lend itself particularly well to the standard format of "introduction-methods-results-conclusion-discussion", but right now (one sentence intro, one sentence discussion, rest results with method and conclusion implied) it is a bit hard to get a good overview of the study.

- Major comments: Pg 3 l13 – pg 5 l15:

Please weigh results to the contribution of the MHC region has made to them. The MHC region is highly variable genetically, a variation that is in no proportion to the variation in the rest of the genome. Functionally the relative impact of this variation is badly understood. Hence the regular practice in interpreting heritable impact is to report the MHC effect separately from that of the rest of the genomic variation.

We thank the Reviewer and agree that genetic variation in the MHC region is important to liability for MS and IBDs. However, we note that some genetic analyses of the MHC can be challenging using currently available statistical methods, due to the complex and long-range LD in this region. For example, LDSC, bivariate LDSC (for estimation of genetic correlation) and S-LDSC (for estimation of tissue- and cell type-specific SNP heritability enrichment) are all sensitive to the complicated LD in the MHC, and it is recommended (by the developers) that the MHC be excluded from these analyses. This is also the case for SMR. For these reasons, many (but not all) of the analyses in our original submission either excluded the MHC region or included the MHC region as a form of sensitivity analysis (e.g. MR analyses with and without MHC SNPs). Exceptions include MTAG and γ -HESS, which we implemented with MHC SNPs included. We note that the developers of MTAG1 and γ -HESS2 do not explicitly refer to the MHC region, but some studies³ implementing these methods chose to exclude MHC SNPs when performing γ -HESS.

In response to the Reviewer's comment, our updated manuscript now includes additional SMR results including MHC SNPs, with the caveat that many fail the HEIDI test, and with the outcome that

this approach failed to identify any MHC genes associated with both MS and IBD, UC or CD (page 20, line 22-26; page 8, line 12-14). We also implemented MAGMA for tissue- and cell type- specific enrichment analyses including the MHC (Methods and Results are described in the Supplementary Notes). These analyses revealed slightly higher heritability enrichments with the inclusion of genes in the MHC region (Table S14-15), suggesting a moderate contribution of the MHC region to shared liability of MS and IBDs. Collectively, our analyses are consistent with a moderate contribution of genetic variation in the MHC region to the shared genetics underlying MS and IBDs. Nevertheless, quantifying the genetic contribution of the MHC region in disease is challenging and further studies are warranted. We have updated the Discussion to reflect these points (page 12, line 4-12).

-- The clarification of the authors solves my question regarding the influence of the MHC region on the results. Mentioning that the MHC region was excluded as a viable source of genetic risk might have been sufficient to answer my question, however I appreciate the extra analyses the authors performed. I think that addressing the influence of the MHC region in a study exploring the overlap in genetic background of three disease phenotypes with known significant MHC genetic risk variants is relevant.

- Pg 6 | 11-28: Very narrow reference of tissues, which makes these results hard to interpret in the wider scope of disease biology.

We thank the Reviewer for this comment. To our knowledge, GTEx remains the largest publicly available resource of tissue-level gene expression data, with 53 tissues collected from approximately 700 non-diseased human donors. As noted in our response to Reviewer #2 (p6 and Figure 4), we performed conditional S-LDSC to correct for potential gene overlap among tissues and found that heritability enrichment for lung, spleen, small intestine and whole blood in MS and IBDs remained significant. These analyses (see details above) strengthen the evidence in our manuscript for SNP heritability enrichments specifically in immune system-related tissues.

-- I agree with the authors that the analysis is simply limited by the availability of reference sets, I don't suggest any other solutions, just to address this limitation in the discussion.

Reviewer #4 (Remarks to the Author):

I appreciate the authors' comprehensive responses and their incorporating in the revised manuscript. These have satisfactorily addressed most of my comments.

Thank you for the clarifications regarding the MTAG methods. That said, I still have concerns over the validity of the cross-trait GWAS meta-analysis given that the assumptions of equal SNP heritability for each trait and perfect genetic covariance between traits do not hold. The developers specify that using these options implies that the GWAS meta-analyzed are considered to be the "same measure of a single trait", which is not the case here. The other studies referenced by the authors in their rebuttal (van der Harst et al 2018, Han et al. 2019) applied this method to meta-analyze GWAS summary statistics of the same phenotype (CAD and optic disc diameter respectively). This is at the very least an important limitation of the cross-trait analysis, and severely undermines the confidence in the reported SNPs from that analysis.

I have no further comments on the remaining sections.

Reviewer 2

The authors made a great work at answering my concerns, especially the ones regarding paper clarity and S-LDSC analyses. I am particularly impressed by the suggested conditional analyses, showing distinct tissues in CD and UC.

We are grateful to the Reviewer for taking the time to review our revised manuscript, and for their positive assessment of our revision.

Really minor points:

- page 4 -line 17: what do you mean by baseline ldsc. Do you mean mean stratified ldsc? (as it seems that you use the baseline-LD model to estimate heritability)

We thank the Reviewer for bringing this to our attention and apologise for the use of imprecise terminology. The Reviewer is correct and we have now revised the Results (page 4; line 17-18) and Methods (page 15; line 14) sections to read: “stratified linkage disequilibrium (LD) score regression (S-LDSC) with the baseline-LD model”.

- page 6 - line 28: whole blood looks also significant for CD in Table S12.

We thank the Reviewer for this suggestion. However, we wish to note that this paragraph is devoted to reporting tissues with FDR-significant enrichments in MS, IBD, UC and/or CD using the conditional S-LDSC model (i.e. adjusted for the baseline model, the set of all genes, and the set of genes specifically expressed in the three non-focal tissues [e.g. small intestine - terminal ileum, lung and spleen in analyses of whole blood]). In relation to heritability enrichment for CD in whole blood, the result was non-significant (FDR>5%) in the conditional S-LDSC analysis (see Figure 4 & Table S12).

Reviewer 3

The changes that the authors made to the manuscript have greatly improved the readability, especially for non-bioinformaticians. All the issues I touched upon in my comments have been largely addressed to my satisfaction. I would like to comment to the authors on two of these (see below).

We thank the Reviewer for taking the time to review our revised manuscript, and for their positive assessment of our revision.

In addition, I would like to suggest to the authors to maybe limit the number of results described in the abstract. I understand that the study does not lend itself particularly well to the standard format of “introduction-methods-results-conclusion-discussion”, but right now (one sentence intro, one sentence discussion, rest results with method and conclusion implied) it is a bit hard to get a good overview of the study.

We thank the Reviewer for this constructive suggestion. We have updated our Abstract to provide a more succinct summary of the results. The revised Abstract now reads (page 2):

“An epidemiological association between multiple sclerosis (MS) and inflammatory bowel disease (IBD) is well-established, but whether this reflects a shared genetic aetiology, and whether consistent genetic relationships exist between MS and the two predominant subtypes of IBD, ulcerative colitis (UC) and Crohn’s disease (CD), remains unclear. Here, we used large-scale genome-wide association study (GWAS) summary data to investigate the shared genetic architecture between MS and IBDs overall and UC and CD independently. We applied cross-trait linkage disequilibrium score regression (LDSC), finding that the genetic correlation between MS and UC was significantly greater than that between MS and CD. On the basis of these genetic correlations, we performed cross-trait meta-analysis for MS and each of IBD, UC and CD, identifying a total of three novel SNPs shared between MS and IBD (rs13428812), UC (rs116555563), and CD (rs13428812, rs9977672). We then used multiple Mendelian randomization methods, finding suggestive but inconclusive evidence for a causal effect of MS on UC and IBD, and no or weak and inconsistent evidence for a causal effect of IBD or UC on MS. There was no evidence for causality in bidirectional analyses of MS and CD. We also investigated tissue- and cell-type-specific enrichment of SNP heritability for each disease using stratified LDSC. At the tissue level, we observed largely consistent patterns of enrichment for MS and IBDs in immune system-related tissues, including lung, spleen, and whole blood, and in contrast to prior studies, small intestine. At the cell-type level, we identified significant enrichment for MS and IBDs in CD4+ T cells in lung and CD8+ cytotoxic T cells in both lung and spleen. These results were largely robust to conditioning on related tissues and cell types. Our study sheds new light on the biological basis of comorbidity between MS and both UC and CD.”

Major comments: Pg 3 l13 – pg 5 l15:

Please weigh results to the contribution of the MHC region has made to them. The MHC region is highly variable genetically, a variation that is in no proportion to the variation in the rest of the genome. Functionally the relative impact of this variation is

badly understood. Hence the regular practice in interpreting heritable impact is to report the MHC effect separately from that of the rest of the genomic variation.

We thank the Reviewer and agree that genetic variation in the MHC region is important to liability for MS and IBDs. However, we note that some genetic analyses of the MHC can be challenging using currently available statistical methods, due to the complex and long-range LD in this region. For example, LDSC, bivariate LDSC (for estimation of genetic correlation) and S-LDSC (for estimation of tissue- and cell type-specific SNP heritability enrichment) are all sensitive to the complicated LD in the MHC, and it is recommended (by the developers) that the MHC be excluded from these analyses. This is also the case for SMR. For these reasons, many (but not all) of the analyses in our original submission either excluded the MHC region or included the MHC region as a form of sensitivity analysis (e.g. MR analyses with and without MHC SNPs). Exceptions include MTAG and \tilde{n} -HESS, which we implemented with MHC SNPs included. We note that the developers of MTAG1 and \tilde{n} -HESS2 do not explicitly refer to the MHC region, but some studies³ implementing these methods chose to exclude MHC SNPs when performing \tilde{n} -HESS. In response to the Reviewer's comment, our updated manuscript now includes additional SMR results including MHC SNPs, with the caveat that many fail the HEIDI test, and with the outcome that this approach failed to identify any MHC genes associated with both MS and IBD, UC or CD (page 20, line 22-26; page 8, line 12-14). We also implemented MAGMA for tissue- and cell type-specific enrichment analyses including the MHC (Methods and Results are described in the Supplementary Notes). These analyses revealed slightly higher heritability enrichments with the inclusion of genes in the MHC region (Table S14-15), suggesting a moderate contribution of the MHC region to shared liability of MS and IBDs. Collectively, our analyses are consistent with a moderate contribution of genetic variation in the MHC region to the shared genetics underlying MS and IBDs. Nevertheless, quantifying the genetic contribution of the MHC region in disease is challenging and further studies are warranted. We have updated the Discussion to reflect these points (page 12, line 4-12).

-- The clarification of the authors solves my question regarding the influence of the MHC region on the results. Mentioning that the MHC region was excluded as a viable source of genetic risk might have been sufficient to answer my question, however I appreciate the extra analyses the authors performed. I think that addressing the influence of the MHC region in a study exploring the overlap in genetic background of three disease phenotypes with known significant MHC genetic risk variants is relevant.

Pg 6 | 11-28: Very narrow reference of tissues, which makes these results hard to interpret in the wider scope of disease biology.

We thank the Reviewer for this comment. To our knowledge, GTEx remains the largest publicly available resource of tissue-level gene expression data, with 53 tissues collected from approximately 700 non-diseased human donors. As noted in our response to Reviewer #2 (p6 and Figure 4), we performed conditional S-LDSC to correct for potential gene overlap among tissues and found that heritability enrichment for lung, spleen, small intestine and whole blood in MS and IBDs remained significant. These analyses (see details above)

strengthen the evidence in our manuscript for SNP heritability enrichments specifically in immune system-related tissues.

-- I agree with the authors that the analysis is simply limited by the availability of reference sets, I don't suggest any other solutions, just to address this limitation in the discussion.

We thank the Reviewer for this positive feedback on our response to their original comments, and for the suggestion to include availability of reference tissue-level gene expression data as a limitation in the discussion (Page 13 line 16-19). This section now reads:

“Lastly, although GTEx remains the largest publicly available resource of tissue-level gene expression data, we acknowledge that larger and more comprehensive tissue-level gene expression resources, including for immune system-related tissues, would enable more detailed investigations of disease biology.”

Reviewer 4

I appreciate the authors' comprehensive responses and their incorporating in the revised manuscript. These have satisfactorily addressed most of my comments.

We thank the Reviewer for taking the time to review our revised manuscript, and for their positive assessment of our revision.

Thank you for the clarifications regarding the MTAG methods. That said, I still have concerns over the validity of the cross-trait GWAS meta-analysis given that the assumptions of equal SNP heritability for each trait and perfect genetic covariance between traits do not hold. The developers specify that using these options implies that the GWAS meta-analyzed are considered to be the "same measure of a single trait", which is not the case here. The other studies referenced by the authors in their rebuttal (van der Harst et al 2018, Han et al. 2019) applied this method to meta-analyze GWAS summary statistics of the same phenotype (CAD and optic disc diameter respectively). This is at the very least an important limitation of the cross-trait analysis, and severely undermines the confidence in the reported SNPs from that analysis.

We thank the Reviewer for pointing out this limitation of our cross-trait GWAS meta-analyses. We acknowledge that the MTAG assumptions of equal SNP heritability for each trait and perfect genetic covariance between traits are violated in our analyses. To determine the extent to which this influences our MTAG results, we performed sensitivity analyses using an alternative cross-trait meta-analysis approach called CPASSOC (Cross Phenotype Association; Zhu *et al.* 2015, *Am J Hum Genet* **96**, 21-36). CPASSOC is a widely cited method (197 cites, Google Scholar May 21st 2021) that assumes the presence of heterogeneous effects across traits and performs a sample size-weighted cross-trait meta-analysis of GWAS summary data. We observed highly consistent results between CPASSOC and MTAG for SNPs reported to be genome-wide significant by MTAG (see newly added Figure S5 below), suggesting that any bias arising from violation of assumptions in MTAG is likely to be minor. Nonetheless, to further allay the reviewer's concerns, we now report only those novel cross-trait SNPs that surpassed genome-wide significance ($p < 5 \times 10^{-8}$) using both MTAG and CPASSOC.

We have updated the manuscript to describe these new CPASSOC analyses in the Abstract (page 2), Methods (page 16, line 32 to page 17, line 4) and Results (page 5, line 15-29). We have additionally updated Table S3, added Figure S5 and edited the Discussion (page 9, line 8-12; page 13, line 2-5) to acknowledge the limitation of our cross-trait GWAS meta-analyses.

The revised sections now read:

Abstract (page 2):

“On the basis of these genetic correlations, we performed cross-trait meta-analysis for MS and each of IBD, UC and CD, identifying a total of three novel SNPs shared between MS and IBD (rs13428812), UC (rs11655563), and CD (rs13428812, rs9977672).”

Methods (page 16, line 32 to page 17, line 5):

“To investigate if violations of the assumptions of equal SNP heritability for each trait and perfect genetic covariance between traits biased our MTAG results, we performed CPASSOC (Cross Phenotype Association) for MS-IBD, MS-UC, and MS-CD as a sensitivity analysis. CPASSOC assumes the presence of heterogeneous effects across traits and estimates the cross-trait statistic S_{Het} and p-value through a sample size-weighted meta-analysis of GWAS summary data. We prioritised independent SNPs that were genome-wide significant in the cross-trait meta-analyses (e.g. MS-IBD) using both MTAG and CPASSOC, but not identified in the original single-trait GWAS (e.g. MS or IBD).”

Results (page 5, line 15-29):

“Based on evidence for significant genetic correlations between MS and each of IBD, UC and CD, we performed cross-trait meta-analyses to identify novel SNPs underlying the joint phenotypes MS-IBD, MS-UC and MS-CD. We used two complementary approaches – MTAG (Multi-Trait Analysis of GWAS)²⁷ and CPASSOC (Cross Phenotype Association)²⁸ – and conservatively prioritised only those SNPs surpassing genome-wide significance ($p < 5 \times 10^{-8}$) using both methods. After excluding SNPs that were genome-wide significant in the respective single-trait GWAS (IMSGC GWAS discovery cohort [14,802 cases, 26,703 controls]⁹; IBD¹²; UC¹²; CD¹²) or the IMSGC GWAS meta-analysis of MS (discovery + replicate cohorts [47,429 cases, 68,374 controls]; $n=200$ non-MHC genome-wide significant SNPs)⁹ or that were in LD ($LD r^2 \geq 0.05$) with any of these previously reported genome-wide significant SNPs, we identified one novel SNP (rs13428812) associated with the joint phenotype MS-IBD, which was also significant in the MS-CD cross-trait GWAS (Table S3). A further two novel SNPs were uniquely associated in the cross-trait GWAS meta-analyses of MS-UC (rs11655563) and MS-CD (rs9977672), respectively. The maxFDR (i.e. the upper bound for the false discovery rate [FDR]) values for MTAG analyses of MS and each of IBD, UC, and CD were roughly 4.55×10^{-7} . Additionally, MTAG results were highly consistent with those generated by CPASSOC (Figure S5), suggesting any bias due to violation of MTAG assumptions is likely to be negligible.”

Discussion (page 9, line 8-12):

“Cross-trait GWAS meta-analyses identified three novel SNPs shared between MS and IBD, UC, and CD. Notably, all three SNPs showed consistent direction of effect between the cross-trait GWAS meta-analyses and the component single trait GWAS’s (Table S3). Moreover, the novel SNP shared by MS-IBD and MS-CD (i.e. rs13428812) also showed the same direction of effect in both cross-trait meta-analyses (Table S3).”

Discussion (page 13, line 2-5):

“First, our cross-trait GWAS meta-analysis results may be biased due to violations of the MTAG assumptions of equal SNP heritability for each trait and perfect genetic covariance between traits. However, such impacts were likely negligible since we performed CPASSOC as a sensitivity analysis and observed highly consistent results.”

Figure S5. Comparison of negative log₁₀ p-values for cross-trait GWAS meta-analyses of MS-IBD, MS-UC and MS-CD performed using MTAG and CPASSOC. Results are displayed for MTAG (x axis) and CPASSOC (y axis) for SNPs with genome-wide significant support ($p < 5 \times 10^{-8}$) in the MTAG analyses, but not in the original single-trait GWAS. Grey: MS-IBD SNPs; correlation=0.98 (95% confidence interval [CI]=0.98-0.99). Orange: MS-UC SNPs; correlation=0.93 (95% CI=0.91-0.94). Blue: MS-CD SNPs; correlation=0.93 (95% CI=0.92-0.94).

REVIEWERS' COMMENTS

Reviewer #4 (Remarks to the Author):

The authors have addressed my remaining concerns. I have no additional comments.